# Duality symmetry, zero energy modes and boundary spectrum of the sine-Gordon/massive Thirring model

Parameshwar R. Pasnoori,[1,2] Ari M. Mizel,[2] and Patrick Azaria[3]

[1]*Department of Physics, University of Maryland,*
*College Park, MD 20742, United States of America*
[2]*Laboratory for Physical Sciences, 8050 Greenmead Dr,*
*College Park, MD 20740, United States of America*
[3]*Laboratoire de Physique Théorique de la Matière Condensée,*
*Sorbonne Université and CNRS, 4 Place Jussieu, 75252 Paris, France*

We solve the one-dimensional massive Thirring model, which is equivalent to the one-dimensional sine-Gordon model, with two types of Dirchlet boundary conditions: open boundary conditions (OBC) and twisted open boundary conditions ($\widehat{\text{OBC}}$). The system exhibits a duality symmetry which relates models with opposite bare mass parameters and boundary conditions, i.e: $m_0 \leftrightarrow -m_0$, OBC $\leftrightarrow \widehat{\text{OBC}}$. For $m_0 < 0$ and OBC, the system is in a trivial phase whose ground state is unique, as in the case of periodic boundary conditions. In contrast, for $m_0 < 0$ and $\widehat{\text{OBC}}$, the system is in a topological phase characterized by the existence of zero energy modes (ZEMs) localized at each boundary. As dictated by the duality symmetry, for $m_0 > 0$ and $\widehat{\text{OBC}}$, the trivial phase occurs, whereas the topological phase occurs for $m_0 > 0$ and OBC. In addition, we analyze the structure of the boundary excitations, finding significant differences between the attractive ($g > 0$) and the repulsive ($g < 0$) regimes.

## I. INTRODUCTION

The boundaries of a quantum system can house localized zero energy modes (ZEM) that play a crucial role in the low-energy physics of the system. Sometimes, this arises from spontaneous symmetry breaking, such as in the case of the antiferromagnetic XXZ spin chain in which the edge zero mode eigenvalues label the two symmetry broken states [1, 2]. However, more commonly, such ZEMs are associated with a topological phase where the symmetry remains unbroken [3–12]. In this case, the edge states cannot be described by the onset of a local order parameter. The resulting symmetry protected ground state degeneracy is the hallmark of a symmetry protected topological (SPT) phase in one dimension [13–21]. Paradigmatic examples of SPT phases are the spin-1 Haldane chain [22, 23] and one dimensional charge-conserving superconductors [7, 24, 25]. In this paper, we focus on the latter case.

In these systems, superconductivity arises as a consequence of the opening of a spin gap due to intrinsic attractive charge-conserving interactions. The charge degrees of freedom remain massless which leads to quasi-long range superconducting correlations decaying as a power law. They are described by the Luttinger liquid model. The spin degrees of freedom are described by the sine-Gordon (SG) model with Hamiltonian $H = \int_0^L dx \, \mathcal{H}(x)$ and

$$\mathcal{H} = \frac{1}{2}[(\partial_x \Phi(x))^2 + (\partial_x \Theta(x))^2] - \lambda \cos \beta \Phi(x). \quad (1)$$

Here, $\Theta(x)$ is the dual of the $\Phi(x)$ field satisfying $[\Theta(x), \Phi(y)] = iY(x - y)$ with $Y(u)$ the Heaviside function. When $\lambda > 0$, the SG model (1) describes spin-singlet superconductors (SSS), whereas when $\lambda < 0$ it

describes SPT spin-triplet superconductors (STS).

The models with $\lambda > 0$ and $\lambda < 0$ differ by their edge properties. Consider imposing the boundary conditions

$$\Phi(0) = 0, \ \Phi(L) = 0 \text{ modulo } 2\pi \quad (2)$$

on Eq. (1). In the semi-classical limit $\beta \to 0$, the SG model with $\lambda < 0$ hosts ZEMs at its two edges, as shown in [7]. It was later shown [24] that these ZEMs survive quantum fluctuations in the weak-coupling limit $\beta \to \sqrt{8\pi}$, where the SG model (1) is equivalent to the spin degrees of freedom of the $U(1)$ Thirring model. These results show that the SPT phase exhibited by STS occurs in both the extreme semi-classical and weak-coupling quantum limits, i.e. $\beta \to 0$ and $\beta \to \sqrt{8\pi}$ respectively. This naturally leads to an important question whether the ZEMs, and hence the topological phase, occur at generic values of $\beta$. In the present work, we show that they do.

In the following, we provide the exact solution of the SG model (1) with boundary conditions (2) for all values $0 \leq \beta \leq \sqrt{8\pi}$ and $\lambda$. To accomplish this, we make use of the equivalence between the SG model (1) and the massive Thirring (MT) model [26]. Using the bosonization formula

$$\Psi_{L,R}(x) = \frac{1}{\sqrt{2\pi a_0}} e^{\mp i\sqrt{\pi}(\Phi(x) \pm \Theta(x))}, \quad (3)$$

where $a_0$ is a short distance cutoff, (1) becomes the MT Hamiltonian density

$$\mathcal{H} = -i(\Psi_R^\dagger(x)\partial_x \Psi_R(x) - \Psi_L^\dagger(x)\partial_x \Psi_L(x))$$
$$+ im_0 \left( \Psi_L^\dagger(x)\Psi_R(x) - \Psi_R^\dagger(x)\Psi_L(x) \right)$$
$$+ 2g \, \Psi_R^\dagger(x)\Psi_L^\dagger(x)\Psi_L(x)\Psi_R(x), \quad (4)$$

where $m_0 = -\pi a_0 \lambda$ and $\beta^2/4\pi = 1 + g/\pi$. Since the SG model describes SSS and STS phases when $\lambda > 0$ and $\lambda < 0$ respectively, the MT model describes SSS and STS phases when $m_0 < 0$ and $m_0 > 0$ respectively. When $g = 0$ the MT model is that of free massive fermions. When $g < 0$ particles and holes attract whereas when $g > 0$ they repel. These correspond to the attractive $\beta^2/4\pi < 1$ and repulsive $\beta^2/4\pi > 1$ regimes of the SG model.

The MT model has been studied extensively [26–31] and has been solved using Bethe ansatz [32] with periodic boundary conditions. In pioneering work, it was also analyzed using Bethe ansatz for the case of open boundary conditions [33]. This study investigated the boundary bound state structure but did not identify the ground states of the system. Here, we construct the ground states of the model with Dirichlet boundary conditions in both the attractive and repulsive regimes using Bethe ansatz. We show that, similar to the weak-coupling regime [25, 34], the model exhibits duality and ZEMs. We then obtain the complete boundary spectrum and show that it significantly depends on the interaction strength $g$.

The paper is structured as follows. In Sec. II, the symmetries of the MT model are discussed. In Sec. III, the Bethe ansatz wavefunction is provided, along with the Bethe equations. The Bethe equations are then solved. The trivial phase is described in Sec. IV. The topological phase is presented in Section V. We conclude in Sec. VI and consider some open questions.

## II. SYMMETRIES AND PROPERTIES

In this section, we recall some of the basic properties of the MT model and discuss the boundary conditions we shall consider.

### A. Periodic boundary conditions

The Hamiltonian (4) is invariant under the $U(1)$ symmetry that conserves the number of particles, or charge,

$$N = \int_0^L dx \; \left( \Psi_L^\dagger(x)\Psi_L(x) + \Psi_R^\dagger(x)\Psi_R(x) \right). \quad (5)$$

Notice that, unlike in the $U(1)$ Thirring model [35], the chiral symmetry $\Psi_{R,L}(x) \to \Psi_{L,R}(x)$ is explicitly broken due to the presence of the mass term. The Hamiltonian is also invariant under the charge conjugation operation

$$\mathcal{C}\Psi_{L,R}^\dagger(x) = \Psi_{L,R}(x), \quad (6)$$

which has the $\mathbb{Z}_2$ group structure $\{1, \mathcal{C}\}$, $\mathcal{C}^2 = 1$. In the SG model, this transformation acts as $\Phi(x) \to -\Phi(x)$.

The fundamental excitations in the system are solitons which have the same mass $m$ in both the SSS and STS phases. It is given by [32]

$$m = \frac{|m_0|\gamma}{\pi(\gamma - 1)} \tan(\pi\gamma) \, e^{\Lambda(1-\gamma)}, \quad (7)$$

$$\text{where} \quad \gamma = \frac{\pi}{2u}, \;\; u = \frac{\pi}{2} - \tan^{-1}\frac{g}{2},$$

where $u$ is a renormalized coupling and $\Lambda = \log(D/|m_0|)$ where $D$ is an UV cutoff. A restriction $u > \pi/3$ is imposed by the regularization scheme [32]. Contact with the SG model is made through the scaling argument

$$\frac{m}{D} \propto \left( \frac{|m_0|}{D} \right)^{1/(2-\beta^2/4\pi)} \quad (8)$$

which yields $\beta^2/4\pi = 2 - 2u/\pi$, matching Coleman's expression for small $g$: $\beta^2/4\pi = 1 + g/\pi$ [26]. In the scaling limit, the dimensionless bare mass term $|m_0|/D$ goes to zero as the cutoff $D$ is sent to infinity while keeping the renormalized mass m fixed, i.e: $|m_0|/D \propto (m/D)^{2u/\pi}$. In terms of the renormalized coupling $u$ in the MT model, the free fermion point is obtained with $u = \pi/2$, while attractive interactions are described by $u > \pi/2$ and repulsive interactions are described by $u < \pi/2$. In the attractive regime, in addition to solitons the system exhibits breathers, which are bound states of solitons and anti-solitons [32]. The mass of the bulk breather corresponding to a bound state of $l$ solitons and anti-solitons is given by

$$m_{Bl} = 2m \sin(l\pi\xi/2), \;\; l = 1, ..., [1/\xi], \;\; \xi = 2\gamma - 1. \quad (9)$$

Here, [...] denotes the integer part, and the integer $l$ is generally called the length of the breather. Hence, as one moves towards the semi-classical limit $u \to \pi$, breathers of increasing length $l$ are allowed, and simultaneously, the mass associated with the breather goes to zero.

### B. Open boundary conditions

The boundary conditions (2) translate into the following open boundary conditions on the fermion fields

$$\Psi_L(0) = e^{i\phi}\Psi_R(0), \;\; \Psi_L(L) = -e^{-i\phi'}\Psi_R(L). \quad (10)$$

where the boundary phases $\phi$ and $\phi'$ to depend on the renormalized coupling $u$ as follows

$$\text{OBC}: \quad \phi = u - \pi/2, \;\; \phi' = u - \pi/2. \quad (11)$$

Here, we use the abbreviation OBC to refer to this specific choice of open boundary conditions. A naive application of the fermion/boson correspondence at the edges would not include the extra $u - \pi/2$ term in (11). This extra term should be thought of as a kind of boundary anomaly [33] necessary for the solution of the massive Thirring model to be consistent with that of the sine-Gordon model. The boundary conditions (10) conserve

the total charge (5) but seem to break the charge conjugation symmetry (6). However, as discussed in the Appendix, the system is in fact invariant under charge conjugation $\mathcal{C}$.

### C. Duality Symmetry

The system exhibits a duality symmetry $\Omega$

$$\Omega \Psi(x) = \hat{\Psi}(x) \tag{12}$$

$$\hat{\Psi}_L(x) = \Psi_L(x), \qquad \hat{\Psi}_R(x) = -\Psi_R(x). \tag{13}$$

In a system with periodic boundary conditions, the duality $\Omega$ relates models with opposite values of the bare masses $m_0$ and the same $g$: $\mathcal{H}(\Psi, g, m_0) = \mathcal{H}(\hat{\Psi}, g, -m_0)$. In the presence of OBC (11) the duality also changes the boundary conditions to dual ones, i.e: $\phi \to \phi + \pi$ and $\phi' \to \phi' + \pi$ that we shall denote

$$\widehat{\text{OBC}}: \quad \phi = u + \pi/2, \ \ \phi' = u + \pi/2. \tag{14}$$

These are twisted open boundary conditions that correspond to the following boundary conditions in the SG model

$$\Phi(0) = \pi, \ \Phi(L) = \pi. \tag{15}$$

Hence the duality symmetry $\Omega$ provides for an isometry that relates, at all energy scales, the system in the STS phase with $m_0 > 0$ and OBC boundary conditions (11) to the system in the SSS phase with $m_0 < 0$ and $\widehat{\text{OBC}}$ boundary conditions (14) and vice versa, i.e:

$$\mathcal{H}(\Psi, g, m_0, \text{OBC}) = \mathcal{H}(\hat{\Psi}, g, -m_0, \widehat{\text{OBC}}). \tag{16}$$

As we shall see, the above duality will play an important role in the boundary physics.

### III. BETHE ANSATZ EQUATIONS

Since the number of particles $N$ (5) is a good quantum number, one can construct the Bethe wave function labelled by $N$ as follows

$$|N\rangle = \sum_{\alpha_1, \ldots, \alpha_N = L, R} \int dx_1 \ldots dx_N \mathcal{A} \Upsilon^{\alpha_1 \ldots \alpha_N}_{\beta_1 \ldots \beta_N}(x_1, \ldots x_N)$$

$$\Psi^\dagger_{\alpha_1}(x_1) \ldots \Psi^\dagger_{\alpha_N}(x_N) |vac\rangle. \tag{17}$$

Here the $\beta_i$ correspond to rapidities, $\mathcal{A}$ represents antisymmetrization with respect to the indices $\alpha_i$ and $x_i$, while

$$\Upsilon^{\alpha_1 \ldots \alpha_N}_{\beta_1 \ldots \beta_N}(x_1, \ldots x_N)$$

$$= \sum_Q \theta(\{x_{Q(j)}\}) \sum_{\delta_1, \ldots, \delta_N = 1, 2} A^{\delta_1 \ldots \delta_N}_Q \prod_{i=1}^N Y^{\alpha_i}_{\delta_i, \beta_i}(x_i). \tag{18}$$

In this expression, $Q$ denotes a permutation of the position orderings of particles with $\theta(\{x_{Q(j)}\})$ a Heaviside function that vanishes unless $x_{Q(1)} \leq \cdots \leq x_{Q(N)}$. The quantity $A^{\delta_1 \ldots \delta_N}_Q$ is an amplitude, and

$$Y^R_{1\beta}(x) = -ie^{-\beta/2} e^{-im_0 x \sinh \beta},$$
$$Y^R_{2\beta} = ie^{\beta/2} e^{im_0 x \sinh \beta},$$
$$Y^L_{1\beta}(x) = e^{\beta/2} e^{-im_0 x \sinh \beta}, \text{and}$$
$$Y^L_{2\beta} = -e^{-\beta/2} e^{im_0 x \sinh \beta}. \tag{19}$$

Applying the Hamiltonian (4) with $m_0 > 0$ to the $N$-particle wavefunction (19), and imposing the boundary conditions (10), (11), we obtain the following set of constraint equations for the rapidities $\beta_j$, called the Bethe equations:

$$e^{2im_0 L \sinh \beta_i} = \left( \frac{\cosh\left(\frac{1}{2}(\beta_i + iu)\right)}{\cosh\left(\frac{1}{2}(\beta_i - iu)\right)} \right)^2 \prod_{j \neq i, j=1}^N \frac{\sinh\left(\frac{1}{2}(\beta_i - \beta_j + 2iu)\right)}{\sinh\left(\frac{1}{2}(\beta_i - \beta_j - 2iu)\right)} \frac{\sinh\left(\frac{1}{2}(\beta_i + \beta_j + 2iu)\right)}{\sinh\left(\frac{1}{2}(\beta_i + \beta_j - 2iu)\right)}. \tag{20}$$

Each eigenstate of the Hamiltonian corresponds to a unique set of roots $\{\beta_j\}$ which satisfy (20). Due to the presence of open boundary conditions, the Bethe equations are symmetric under the reflection $\beta_i \leftrightarrow -\beta_i$, so that the solutions to (20) come in pairs $(\beta_i, -\beta_i)$. In the thermodynamic limit $N, L \to \infty$, the roots form a dense set, and the Bethe equations can be expressed as integral equations which can be solved using a Fourier transform.

The energy of the thus obtained eigenstate is given by

$$E = \sum_{j=1}^N m_0 \cosh(\beta_j). \tag{21}$$

In the following, we shall present the solutions to the Bethe equations (20), which correspond to $m_0 > 0$ for OBC (11), and also to the Bethe equations which correspond to $m_0 < 0$ with OBC whose explicit form is given in the Appendix. For a given sign of the mass parameter $m_0$, the system exhibits both trivial and topologi-

cal phases depending on the boundary conditions. For $m_0 < 0$, the boundary conditions OBC (11) give rise to the trivial phase and $\widehat{\text{OBC}}$ (14) give rise to the topological phase. Due to the duality symmetry (13), for $m_0 > 0$, the boundary conditions $\widehat{\text{OBC}}$ (14) give rise to the trivial phase whereas the boundary conditions OBC (11) give rise to the topological phase. As we shall show below, in the trivial phase, the system exhibits a unique ground state. In the topological phase, the system exhibits a fourfold degenerate ground state. This occurs due to the existence of ZEMs exponentially localized at the two boundaries. These ZEMs are associated with special solutions of Bethe equations called "short" or "close" boundary strings [24, 25, 36, 37] in the repulsive regime and in the attractive regime for $u < 2\pi/3$. In the attractive regime for $u > 2\pi/3$, in addition to the close boundary strings, the ZEMs are associated with a new type of boundary string, which we name a "breather boundary string."

## IV. TRIVIAL PHASE

We start by describing the trivial phase. We discuss the case $m_0 < 0$ with OBC (11). The situation when $m_0 > 0$ with $\widehat{\text{OBC}}$ (14) is obtained by the duality symmetry. In the trivial phase, the Hamiltonian has a unique ground state whose total charge is

$$N = q \frac{m_0 L e^\Lambda}{2\pi}, \ q = \frac{\pi}{2(\pi - u)}. \tag{22}$$

One obtains its normal ordered charge by removing from (5) the bulk contribution which is the charge associated with a system with periodic boundary conditions (see the Appendix for details). The normal ordered charge of the ground state when expressed in units of the soliton charge $\mathcal{N} =: N : /q$, identifies with the topological charge of the SG model

$$\mathcal{N} = \frac{1}{\sqrt{\pi}} \int_0^L dx \ \partial_x \Phi(x). \tag{23}$$

According to this definition, the ground state has charge zero, and hence it can be labelled as

$$|GS\rangle = |0\rangle. \tag{24}$$

The advantage of working with this definition of the charge operator is that one can define a fermionic parity

$$\mathcal{P} = (-1)^{\mathcal{N}} \tag{25}$$

associated with each state.

## V. TOPOLOGICAL PHASE

For $m_0 < 0$ with $\widehat{\text{OBC}}$ (14), or equivalently for $m_0 > 0$ with OBC (11), the situation changes completely. ZEMs

arise that are exponentially localized at the left and right edges of the system. As a result, we find four degenerate ground states that we shall label according to their normal ordered charges. There exist two ground states with charges $\pm 1$ and two ground states with charge zero, which have odd and even fermionic parities respectively:

$$\{|-1\rangle, |+1\rangle\} \ \mathcal{P} = -1,$$
$$\{|0\rangle_+, |0\rangle_-\} \ \mathcal{P} = +1. \tag{26}$$

One can construct linear combinations of the two charge zero states $|0\rangle_\pm$ to form

$$|0\rangle_{\mathcal{L},\mathcal{R}} = \frac{1}{\sqrt{2}} \left( |0\rangle_+ \pm |0\rangle_- \right). \tag{27}$$

One can then define the local ZEM parities associated with each state:

$$\mathcal{P}_{\mathcal{L}} = (-1)^{n_{\mathcal{L}}}, \ \mathcal{P}_{\mathcal{R}} = (-1)^{n_{\mathcal{R}}}, \tag{28}$$

where $n_{\mathcal{L},\mathcal{R}} = 0, 1$ is the number of ZEMs at the left and right boundaries respectively. This allows us to label the four ground states described above using the local ZEM parities as shown in Table I. In the basis $\{|-1\rangle, |0\rangle_{\mathcal{L}}, |0\rangle_{\mathcal{R}}, |1\rangle\}$, under charge conjugation, the two states in each fermionic parity sector transform into each other.

TABLE I: The four ground states and their respective local ZEM parities $\mathcal{P}_{\mathcal{L},\mathcal{R}}$.

| State | $\mathcal{P}_{\mathcal{L}}$ | $\mathcal{P}_{\mathcal{R}}$ |
|---|---|---|
| $|-1\rangle$ | 1 | 1 |
| $|0\rangle_{\mathcal{L}}$ | -1 | 1 |
| $|0\rangle_{\mathcal{R}}$ | 1 | -1 |
| $|1\rangle$ | -1 | -1 |

The construction of these states differs significantly for different values of the interaction strength. Below we shall summarize the construction of these four ground states in the repulsive $(u < \pi/2)$ regime and in the attractive $(u > \pi/2)$ regime separately.

### A. Repulsive regime

In the Bethe ansatz approach, the ground state $|1\rangle$ is obtained by choosing all Bethe roots to lie on the $i\pi$ line. Charge conjugation invariance of the system implies that there exists another degenerate state with opposite charge, which we label $|-1\rangle$. Within Bethe ansatz, as explained in detail in the Appendix, the state $|-1\rangle$ can be obtained by starting with Bethe equations corresponding to the charge conjugated fermions and then choosing all the roots to lie on the $i\pi$ line. This point is essential in understanding the ground state structure of the system.

The state $|0\rangle_+$ is obtained by adding the fundamental boundary string solution of the Bethe equations to

the state $|1\rangle$ or $|-1\rangle$. This purely imaginary solution arises due to a double pole in the Bethe equations, and corresponds to the rapidity value $\lambda_{(0)} = \pm iu$, where $\lambda = \beta - i(2m+1)\pi$, $m \in \mathbb{Z}$. We refer to this as the "fundamental boundary string solution." In the repulsive regime, this solution falls into the category called close boundary strings [36, 37]. Adding such a close boundary

string results in the new state $|0\rangle_+$ such that the difference of the charge of the original and the final states is equal to that of a soliton or an anti-soliton. See Appendix for more details. In addition, the change in the energy due to the addition of this boundary string is zero in the current regime.

The close boundary strings are associated with exponentially localized wave functions. Since our model possesses space parity symmetry which exchanges the left and the right boundaries $\mathcal{L} \leftrightarrow \mathcal{R}$, the wavefunctions are localized at both boundaries in a symmetric superposition [33]. Due to the double pole associated with the boundary string, there exists a degenerate state, which we label as $|0\rangle_-$, in which the boundary bound state wavefunctions are in an anti-symmetric superposition. [38]. Hence the state $|0\rangle_+$ ($|0\rangle_-$) contains boundary bound states that are localized at both boundaries in a symmetric (anti-symmetric) superposition on top of the state $|1\rangle$ or $|-1\rangle$. These bound states have zero energy – they are ZEMs – and carry charge equal to that of a soliton or anti-soliton. Recall that in the current regime, they are associated with the fundamental boundary string $\lambda_{(0)}$.

## B. Attractive regime

The construction of the states $|\pm 1\rangle$ in the attractive regime exactly parallels that in the repulsive regime, where they are obtained by choosing all the Bethe roots to lie on the $i\pi$ line. The construction of the states $|0\rangle_\pm$ is, however, drastically different. Moreover, a different construction is needed in each regime

$$\frac{2n}{2n+1}\pi < u < \frac{2n+2}{2n+3}\pi, \ n \in \mathbb{N} \qquad (29)$$

where $\mathbb{N}$ denotes the natural numbers including 0.

### 1. $u < 2\pi/3$

For pedagogical clarity, let us start by considering the case $n = 0$. This corresponds to the entire repulsive regime $u < \pi/2$ and the part of the attractive regime with $\pi/2 < u < 2\pi/3$. In this case, the construction of the states $|0\rangle_\pm$ exactly parallels that in the repulsive regime, where $|0\rangle_+$ is obtained by adding the fundamental boundary string $\lambda_{(0)} = \pm iu$ to the state $|1\rangle$ or $|-1\rangle$. As mentioned before, due to the existence of the double

pole, there exists another state, which we label $|0\rangle_-$. Exactly as in the repulsive regime, the ZEM are associated with the fundamental boundary string $\lambda_{(0)}$.

### 2. $2\pi/3 < u < 4\pi/5$

Now consider the case $n = 1$ in (29), which corresponds to $2\pi/3 < u < 4\pi/5$. In this regime, the nature of the fundamental boundary string $\lambda_{(0)}$ changes so that, instead of being a close boundary string, it now becomes what we call a "breather boundary string." Adding a breather boundary string results in a new state, but, unlike the close boundary string, the charge of the final state equals that of the original state. See Appendix for more details. In the current case, adding the breather boundary string $\lambda_{(0)}$ to the state $|1\rangle$ ($|-1\rangle$) results in a state that we label $|1\rangle_{+B0}$ ($|-1\rangle_{+B0}$). Again, the boundary breather wavefunction is localized at both boundaries in a symmetric superposition. Due to the double pole, there exists another state corresponding to an anti-symmetric superposition, which we label as $|1\rangle_{-B0}$ ($|-1\rangle_{-B0}$). The energy difference between the states $|\pm 1\rangle_{\pm,B0}$ and the ground states $|\pm 1\rangle$ is given by

$$E_{B0} = m\sin(\pi\xi), \ \ \xi = 2\gamma - 1, \qquad (30)$$

where $E_{B0} = E_{|\pm 1\rangle_{\pm,B0}} - E_{|\pm 1\rangle}$ and $\gamma$ is defined in (7). Since we are considering $n = 1$ in (29), it follows that $1/2 < \gamma < 1$, implying that the energy of the states $|\pm 1\rangle_{\pm,B0}$ containing the breather boundary string is higher than that of the ground states $|\pm 1\rangle$. Upon choosing one of the roots of the Bethe equations (20) to be the boundary string $\lambda_{(0)} = \pm iu$, a new boundary string $\lambda_{(1)} = \pm 3iu$ solution emerges for $u > 2\pi/3$. Note that this "higher order boundary string" is not a solution to the Bethe equations (20) for $u < 2\pi/3$. This higher order boundary string is a close boundary string in the current regime, and as a result, adding this boundary string to the state $|1\rangle_{+B0}$ ($|-1\rangle_{+B0}$) obtained above, we find that the normal ordered charge of the resulting state is zero. We also find that the energy of this boundary string is exactly equal and opposite to (30). Hence, by adding the higher order boundary string $\lambda_{(1)}$ to the state $|1\rangle_{+B0}$ ($|-1\rangle_{+B0}$) we obtain the ground state $|0\rangle_+$. Again, due to the double pole, there exists another

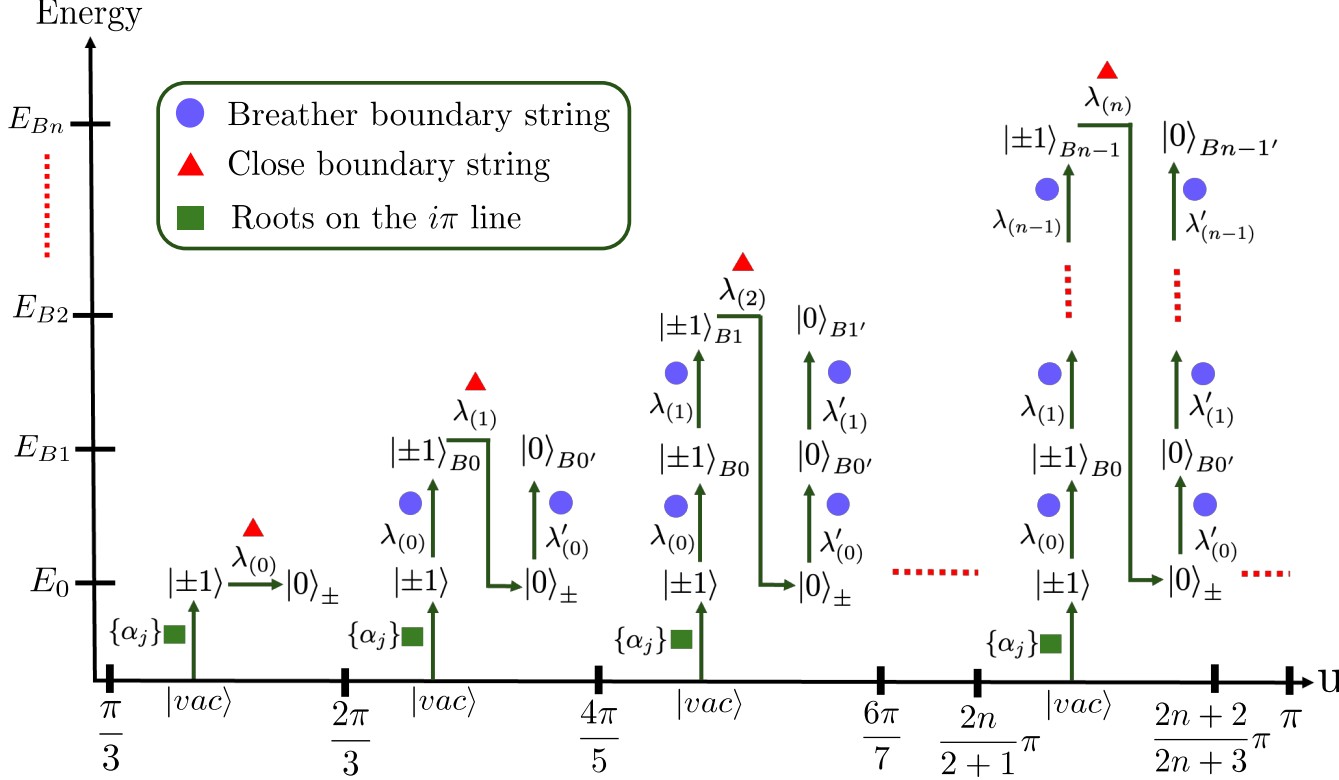

FIG. 1: Figure shows the construction of the four ground states and the boundary excited states for the entire range of allowed values of $u$, which can be partitioned into regimes $\frac{2n}{2n+1}\pi < u < \frac{2n+2}{2n+3}\pi$, $n \in \mathbb{N}$. In each regime, there exist boundary string solutions which fall into two branches. In each branch, there exist $n+1$ boundary string solutions, where each solution corresponds to a double pole in the Bethe equations. The boundary strings corresponding to the first branch are $\lambda_{(j-1)} = \pm i(2j-1)$, $j = 1, 2, ...n+1$. The boundary strings $\lambda_{(j-1)}, j = 1, ...n$ fall into breather boundary string category, whereas the boundary string $\lambda_{(n)}$ falls into the close boundary string category. Green squares represent the continuum of roots on the $i\pi$ line. Red triangles represent close boundary strings and blue circles represent breather boundary strings. The ground states $|\pm1\rangle$ are constructed starting from the vacuum $|vac\rangle$ by adding the dense set of roots $\{\alpha_k\}$ on the $i\pi$ line, which form a distribution $\rho_{|\pm1\rangle}(\alpha)$. The energy of these ground states is represented by $E_0$. By adding $k+1$ breather boundary strings $\lambda_{(j-1)}, j = 1, ..., k+1$ to the states $|\pm1\rangle$, one obtains the states $|\pm1\rangle_{+Bk}$ respectively, where $k \leq n-1$. The total normal ordered charge of these states is equal to that of the states $|\pm1\rangle$ respectively and their energy is given by $E_{Bk} > 0$ (32). Here $+$ in the subscript corresponds to the wavefunction associated with the boundary breathers localized at both the boundaries in a symmetric superposition. Due to the double pole associated with the boundary strings, there exist degenerate states, which are labeled as $|\pm1\rangle_{-Bk}$. To ease the notation in the figure, $\pm$ signs are suppressed in the subscripts of all excited states. By adding all the allowed breather boundary strings along with the close boundary string $\lambda_{(n)}$ to either of the states $|\pm1\rangle$, one obtains the state $|0\rangle_+$. Due to the double poles associated with the boundary strings, there exists another degenerate state $|0\rangle_-$. These state are degenerate with the states $|\pm1\rangle$. This occurs since the energy of the close boundary string $\lambda_{(n)}$ is exactly equal and opposite to the sum of the energies of the breather boundary strings $\lambda_{(j-1)}, j = 1...n$. The boundary strings corresponding to the second branch $\lambda'_{(j-1)}$ take the same form as those in the first branch, and similarly, the first $n$ boundary strings $\lambda'_{(j-1)}, j = 1, ..., n$ are breather boundary strings, whereas the highest order boundary string $\lambda'_{(n)}$ is a close boundary string. By adding $k+1$ breather boundary strings $\lambda'_{(j-1)}, j = 1, ..., k+1$ to the states $|0\rangle_\pm$, one obtains the states $|0\rangle_{\pm,+Bk}$, $k \leq n-1$, whose total normal ordered charge is equal to that of the states $|0\rangle_\pm$ and their energy is given by $E_{Bk} > 0$ (32). In the state $|0\rangle_{\pm,+Bk}$, the wavefunction associated with the boundary breathers localized at both the boundaries is in a symmetric superposition. Again, due to the double pole associated with the boundary strings, there exists a degenerate state corresponding to the anti-symmetric superpositon, which is labeled as $|0\rangle_{\pm,-Bk}$. Hence the states $|0\rangle_{\pm,\pm Bk}$ and $|\pm1\rangle_{\pm Bk}$ which are excited states on top of the four degenerate ground states $|0\rangle_\pm$ and $|\pm1\rangle$ respectively, are also degenerate, and have energy $E_{Bk}$ (32) with respect to the ground states.

degenerate state $|0\rangle_-$. Just as before, the states $|0\rangle_\pm$ contain ZEMs localized at both the boundaries in a symmetric (anti-symmetric) superposition on top of the state

$|1\rangle$ or $|-1\rangle$. But now since the states $|0\rangle_\pm$ are obtained by adding both allowed boundary strings $\lambda_{(0)}$ and $\lambda_{(1)}$ to the state $|1\rangle$ or $|-1\rangle$, unlike in the previous regime,

the ZEMs are associated with the set of the two allowed boundary strings $\lambda_{(0)}$ and $\lambda_{(1)}$.

### 3. $2n\pi/(2n+1) < u < (2n+2)\pi/(2n+3), \ n > 1$

Now consider the general case where $n \in \mathbb{N}$. As mentioned above, there exists a boundary string solution $\lambda_{(0)}$ (fundamental boundary string) that corresponds to the double pole in the Bethe equations. When this is included as one of the roots in the Bethe equations, a new higher order boundary string solution $\lambda_{(1)}$ arises. Then, when $\lambda_{(1)}$ is included in turn, a new higher order boundary string solution $\lambda_{(2)} = \pm 5iu$ emerges. As shown in Fig. 1, his process continues until one reaches the highest order boundary string solution $\lambda_{(n)} = \pm(2n+1)iu$ allowed in the current regime of values of $u$. One can consider the above set of solutions as forming a branch, say the "first branch." In this branch there exist $n+1$ boundary strings

$$\lambda_{(j-1)} = \pm i(2j-1)u, \ j = 1, 2, ..., n+1. \qquad (31)$$

The boundary strings $\lambda_{(j-1)}$, $j = 1, ..., n$ fall into the category of breather boundary strings, whereas the boundary string $\lambda_{(n)}$ falls into the category of close boundary strings. As one increases the values of $u$ and moves from one regime to another, the highest order boundary string corresponding to the current regime $\lambda_{(n)}$ turns into a breather boundary string. For example, for $u < 2\pi/3$, the only allowed boundary string is $\lambda_{(0)}$, which is a close boundary string. As one moves into the regime $2\pi/3 < u < 4\pi/5$, the boundary string $\lambda_{(0)}$ becomes a breather boundary string, and a new boundary string $\lambda_{(1)}$ appears, which is a close boundary string in this regime. As we continue increasing $u$ into the regime $4\pi/5 < u < 6\pi/7$, the boundary string $\lambda_{(0)}$ remains as the breather boundary string, and the boundary string $\lambda_{(1)}$, which was a close boundary string now turns into a breather boundary string. Simultaneously, a new higher order boundary string $\lambda_{(2)}$ appears, which is a close boundary string. This repeats indefinitely as one approaches the semi-classical limit $u \to \pi$ ($n \to \infty$).

The states $|\pm 1\rangle$ are constructed starting from the vacuum $|vac\rangle$ by adding the dense set of roots $\{\alpha_k\}$ on the $i\pi$ line, which form a distribution $\rho_{|\pm 1\rangle}(\alpha)$. By adding $k+1$ breather boundary strings $\lambda_{(j-1)}$, $j = 1, ..., k+1 < n$ to the state $|1\rangle$ ($|-1\rangle$), one obtains the state $|1\rangle_{+Bk}$ ($|-1\rangle_{+Bk}$). Again, due to the double pole structure, there exists another state $|1\rangle_{-Bk}$ ($|-1\rangle_{-Bk}$). The total normal ordered charge of $|1\rangle_{\pm Bk}$ ($|-1\rangle_{\pm Bk}$) is equal to that of $|1\rangle$ ($|-1\rangle$), and its energy is given by

$$E_{Bk} = m \sin(k\pi\xi), \ \ k = 1, ..., n, \ \ n = [1/2\xi], \qquad (32)$$

where $E_{Bk} = E_{|\pm 1\rangle_{\pm, Bk}} - E_{|\pm 1\rangle}$. As in (9), [...] denotes the integer part. We find that the energy of a boundary breather $E_{Bk}$ (32) is always smaller than the mass of a bulk breather $m_{Bk}$ (9). This is depicted in Fig. 2.

By adding all the allowed $n$ breather boundary strings $\lambda_{(j-1)}$, $j = 1, ..., n$ to the state $|\pm 1\rangle$, one obtains the state $|1\rangle_{+Bn-1}$ ($|-1\rangle_{+Bn-1}$), which also has the same charge as that of the state $|1\rangle$ ($|-1\rangle$). By adding the last boundary string $\lambda_{(n)}$, which is the close boundary string, to the state $|1\rangle_{+Bn-1}$ ($|-1\rangle_{+Bn-1}$), one obtains a state whose total normal ordered charge is zero. The energy of the close boundary string is given by

$$E_{CBSn} = -m \sin(n\pi\xi), \qquad (33)$$

which is exactly equal and opposite to $E_{Bn}$, the sum of the energies of the breather boundary strings $\lambda_{(j-1)}$, $j = 1, ..., n$. Hence, by adding the last boundary string $\lambda_{(n)}$ to the state $|1\rangle_{+Bn-1}$ ($|-1\rangle_{+Bn-1}$), we obtain the state $|0\rangle_{+}$. Again, due to the double pole structure, there exists another degenerate state $|0\rangle_{-}$. Just as before, the states $|0\rangle_{+}$ ($|0\rangle_{-}$) contain ZEMs localized at both boundaries in a symmetric (anti-symmetric) superposition. But now since the states $|0\rangle_{\pm}$ are obtained by adding all the allowed $n+1$ boundary strings $\lambda_{(j-1)}$, $j = 1, ..., n+1$ of the first branch to the state $|1\rangle$ or $|-1\rangle$, the ZEMs are associated with the entire set of boundary strings corresponding to the first branch.

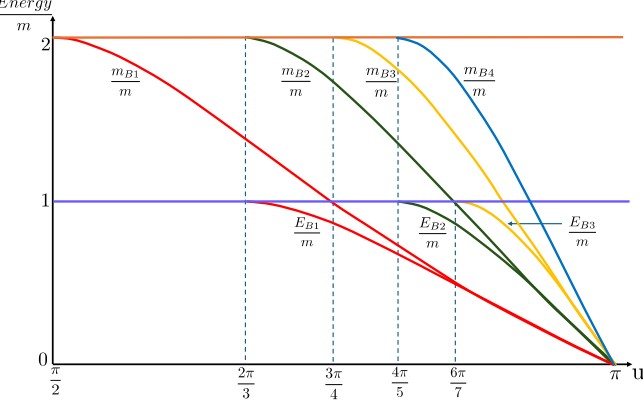

FIG. 2: Figure shows the energies of the first three boundary breathers $E_{B1}, E_{B2}, E_{B3}$ and the masses of the first four bulk breathers $m_{B1}, m_{B2}, m_{B3}, m_{B4}$ in units of the mass of the soliton $m$. The energy of the $n^{th}$ boundary breather is always less than the mass of the $n^{th}$ bulk breather, and, in the semi-classical limit $u \to \pi$, they asymptotically take the same value. In this limit, their functional dependence on $u$ takes the form $m(\pi - u)$.

In the process of constructing the ground states $|0\rangle_{\pm}$, we obtained the excited states $|\pm 1\rangle_{\pm, Bk}$, $k = 0, ..., n-1$ at the boundaries on top of the ground states $|\pm 1\rangle$. Similarly, there exist excited states at the boundaries on top of the ground states $|0\rangle_{\pm}$, whose construction we now describe.

As discussed above, for a given $n$ in (29), there exist $n+1$ boundary strings $\lambda_{(j-1)} = \pm i(2j-1)u$, $j = 1, 2, ..., n+1$. The boundary strings $\lambda_{(j-1)}$, $j = 1, ..., n$ fall into the breather boundary string category, whereas the boundary string $\lambda_{(n)}$ falls into the close boundary

string category. When the fundamental boundary string $\lambda_{(0)}$ is considered as a root, in addition to the higher order boundary string $\lambda_{(1)}$ mentioned above, the Bethe equations give rise to another solution $\lambda'_{(0)}$, which has the same form as $\lambda_{(0)}$. As explained in the Appendix, although $\lambda'_{(0)}$ takes the same form as $\lambda_{(0)}$, it is a new solution distinct from the fundamental boundary string, and hence it is labeled with a prime. By including $\lambda'_{(0)}$ as one of the roots, a new higher order boundary string solution $\lambda'_{(1)}$ arises, which takes the same form as $\lambda_{(1)}$. Again, this is a distinct solution and is hence labeled with a prime. This process continues and one finds that there exist $n+1$ boundary strings $\lambda'_{(j-1)}$, $j=1,...n+1$ which form a new branch, say the "second branch." Since they take the same form as the boundary strings corresponding to the first branch, they have the same charge and energy. Similar to the boundary strings corresponding to the first branch, the first $n$ boundary strings $\lambda'_{(j-1)}$, $j=1,...,n$ are breather boundary strings whereas the highest order boundary string $\lambda'_{(n)}$ is a close boundary string.

The excited states on top of the ground states $|0\rangle_\pm$ are constructed by adding the boundary strings corresponding to the second branch. By adding $k+1$ breather boundary strings $\lambda'_{(j-1)}$, $j=1,...,k+1 < n$ to the state $|0\rangle_+$ ($|0\rangle_-$), one obtains the state $|0\rangle_{+,+Bk}$ ($|0\rangle_{-,+Bk}$), whose total normal ordered charge is equal to that of the states $|0\rangle_\pm$ and its energy with respect to the ground states is given by $E_{Bk}$ (32). Note that the wavefunctions associated with the boundary breathers in the states $|0\rangle_{+,+Bk}$ ($|0\rangle_{-,+Bk}$) are localized at both of the boundaries in a symmetric superposition. Due to the double pole, there exist degenerate states corresponding to an anti-symmetric superposition, which we label $|0\rangle_{+,-Bk}$ ($|0\rangle_{-,-Bk}$). Since the states $|0\rangle_\pm$ are degenerate with the states $|\pm1\rangle$, we see that the excited states $|0\rangle_{\pm,\pm Bk}$ and $|\pm1\rangle_{\pm Bk}$ are also degenerate. By adding all the allowed $n$ breather boundary strings to the state $|0\rangle_+$ ($|0\rangle_-$), one obtains the boundary excited state $|0\rangle_{+,+Bn-1}$ ($|0\rangle_{-,+Bn-1}$), which has the highest energy. Note that unlike the states $|\pm1\rangle_{\pm Bn-1}$, the highest order boundary string $\lambda'_{(n)}$ cannot be added to the state $|0\rangle_{\pm,+Bn-1}$, unless a soliton (hole) is added.

Note that we have constructed the above states by adding only one boundary string corresponding to each double pole. As explained in detail in the appendix, one can add both the boundary strings corresponding to the double poles and obtain the excited states which contain the boundary breathers localized at both the boundaries. Working in the basis where the ground states are represented by $|0\rangle_{L,R}$, the excited states containing the boundary breathers on top of the respective ground states can be represented by $|0\rangle_{L,LBk}$, $|0\rangle_{L,RBk}$ and $|0\rangle_{R,LBk}$, $|0\rangle_{R,RBk}$. Here, the states $|0\rangle_{L,LBk}$, $|0\rangle_{L,RBk}$ contain $k+1$ breather boundary strings corresponding to the left and right edges respectively on top of the ground

state $|0\rangle_L$. Similarly, the states $|0\rangle_{R,LBk}$, $|0\rangle_{R,RBk}$ contain $k+1$ breather boundary strings corresponding to the left and right edges respectively on top of the ground state $|0\rangle_R$. Similarly, the states $|\pm1\rangle_{LBk}$ and $|\pm1\rangle_{RBk}$ contain $k+1$ breather boundary strings corresponding to the left and the right edges respectively on top of the ground states $|\pm1\rangle$. These states have energy $E_{Bk}$ (32). There exist excited states which contain boundary breathers localized at both the boundaries. These states are represented by $|0\rangle_{L,LBk,RBl}$, $|0\rangle_{R,LBk,RBl}$ and $|\pm1\rangle_{LBk,RBl}$, where they correspond to states containing $k+1$ breather boundary strings corresponding to the left boundary and $l+1$ breather boundary strings corresponding to the right boundary on top of the ground states $|0\rangle_{L,R}$ and $|\pm1\rangle$ respectively. These states have energies $E_{Bk} + E_{Bl}$.

## VI. SUMMARY AND OPEN QUESTIONS

In this work, we analyzed the massive Thirring model, which is equivalent to the sine-Gordon model, with two types of Dirichlet boundary conditions. The first type, which we termed open boundary conditions (OBC), is specified in (2) and (11). The second type, called twisted open boundary conditions ($\widehat{\text{OBC}}$), is specified in (14) and (15). Employing Bethe ansatz, we solved the model in both the case of repulsive interactions and the case of attractive interactions. The nature of our solution crucially depends on the sign of the bare fermion mass parameter $m_0$ and the boundary conditions. For $m_0 < 0$, when OBC are applied, the system is shown to exhibit a trivial phase. When $\widehat{\text{OBC}}$ are applied, the system exhibits a topological phase. The system possesses a duality symmetry: $m_0 \leftrightarrow -m_0$, OBC $\leftrightarrow \widehat{\text{OBC}}$, due to which the nature of the phase associated with a given choice of boundary conditions is interchanged when $m_0 > 0$ instead of $m_0 < 0$.

In the trivial phase, the system exhibits a unique ground state. In the topological phase, there exist ZEMs exponentially localized at both boundaries of the system. They have the same charge as the fundamental excitation of the bulk, which is the soliton, but have zero energy. Due to this, the ground state is fourfold degenerate (26), with two states in each fermionic parity sector. The two states transform into each other under charge conjugation and form a representation of the group $\mathbb{Z}_2$. In the repulsive regime, the excitations consist of solitons and bulk breathers which are separated from the ground state by finite energy. In the attractive regime, in addition to solitons and bulk breathers, there exist boundary breathers. Boundary breathers are bound states of solitons and anti-solitons localized at the boundary with finite energy. The number of boundary excitations on top of each of the four ground states is equal. This number increases as the interaction parameter $u$ moves towards the semi-classical limit $u \to \pi$. In addition, in the semi-

classical limit, the energy associated with each boundary breather decreases and asymptotically becomes equal to that of the respective bulk breather; eventually, the energies of both go to zero when $u = \pi$.

At the free fermion point where $u = \pi/2$, in the subspace spanned by the ground states, one defines operators $\mathcal{N}_{\mathcal{L,R}}$, which measure the local charge associated with the edges. These operators have fractional eigenvalues $\pm 1/2$. Hence, the charges associated with the boundaries are sharp quantum observables [39, 40]. Due to this, the $\mathbb{Z}_2$ charge conjugation symmetry fractionalizes between the two edges, i.e. $\mathbb{Z}_2 \implies \mathbb{Z}_{2\mathcal{L}} \otimes \mathbb{Z}_{2\mathcal{R}}$, and there exist two zero-energy Majorana modes exponentially localized at each edge. An important open question is whether these remarkable properties survive interactions away from the free fermion point $u = \pi/2$. Within the Bethe ansatz framework, an interesting route to investigate this issue could be to follow the approach given in [41] for driven quantum circuits.

Another important open question concerns the effect of changes in the values of the boundary fields (11), so that $\phi$ and $\phi'$ differ from $u - \pi/2$. If this difference is small, the ground state degeneracy should be lifted, giving rise to mid-gap states. For larger differences, more profound changes should occur. The nature of these changes is currently under investigation [42].

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

## Appendix A: The Bethe wave function

The Hamiltonian of the Massive Thirring model is

$$H = -i \int_0^L \Psi_R^\dagger(x)\partial_x\Psi_R(x) + i\int_0^L \Psi_L^\dagger(x)\partial_x\Psi_L(x) + im_0\int_0^L \Psi_L^\dagger(x)\Psi_R(x) - \Psi_R^\dagger(x)\Psi_L(x)$$

$$+2g\int_0^L \Psi_R^\dagger(x)\Psi_L^\dagger(x)\Psi_L(x)\Psi_R(x). \tag{A1}$$

Here $\Psi_{L,R}(x)$ are the fermionic fields, $m_0$ is the bare mass and $g$ is the bulk interaction strength. We apply the following boundary conditions on the fermionic fields

$$\Psi_L(0) = e^{i\phi}\Psi_R(0), \quad \Psi_L(L) = -e^{-i\phi'}\Psi_R(L), \tag{A2}$$

where $\phi, \phi'$ are phases. The boundary conditions (2) of the sine-Gordon model translate to the following relations

$$\text{OBC}: \phi = u - \pi/2, \ \ \phi' = u - \pi/2, \tag{A3}$$

and $u = \pi/2 - \tan^{-1}(g/2)$. As discussed in the main text, the system exhibits a duality symmetry that relates the Hamiltonian with mass parameter $m_0$ and open boundary conditions (OBC) to that with mass $-m_0$ and dual open boundary conditions ($\widehat{\text{OBC}}$) (A2):

$$\mathcal{H}(\Psi, m_0, \text{OBC}) = \mathcal{H}(\hat{\Psi}, -m_0, \widehat{\text{OBC}}), \tag{A4}$$

where

$$\widehat{\text{OBC}}: \phi = u + \pi/2, \ \ \phi' = u + \pi/2. \tag{A5}$$

Since the duality transformation (A4) maps the system with $m_0 < 0$ ($m_0 > 0$) and OBC to the system with $m_0 > 0$ ($m_0 < 0$) and $\widehat{\text{OBC}}$, it suffices to solve the Hamiltonian (A1) with $m_0 > 0$ with both OBC and $\widehat{\text{OBC}}$. Below we shall provide the details of the construction of the Bethe wave function and obtain the Bethe equations by applying the appropriate boundary conditions. In Sec. A 3, we solve the Bethe equations and obtain the spectrum associated with the boundaries.

Since the Hamiltonian (A1) conserves the total number of particles $N$, where

$$N = \int_0^L dx \left( \Psi_L^\dagger(x)\Psi_L(x) + \Psi_R^\dagger(x)\Psi_R(x) \right), \tag{A6}$$

we shall construct the Bethe ansatz wave function in each $N$ sector.

### 1. One particle sector

In one particle sector, the wave function can be written as

$$|1\rangle = \int_0^L dx \left( \chi_\beta^R(x)\Psi_R^\dagger(x) + \chi_\beta^L(x)\Psi_L^\dagger(x) \right) |vac\rangle, \tag{A7}$$

where $\beta$ is the rapidity. Applying the Hamiltonian (A1), to the one particle wave function (A7), we obtain the following set of equations

$$(-i\partial_x - E)\chi_\beta^R(x) - im_0\chi_\beta^L = 0,$$
$$(i\partial_x - E)\chi_\beta^L(x) + im_0\chi_\beta^R = 0, \tag{A8}$$

where $E$ is the energy. To solve the above set of equations, we use the following ansatz for $\chi_\beta^{L,R}(x)$:

$$\chi_\beta^R(x) = \mathcal{N}\sum_{i=1}^2 Y_{i\beta}^R(x)A_i(\beta, \phi)\,\theta(x)\theta(L-x), \quad \chi_\beta^L(x) = \mathcal{N}\sum_{i=1}^2 Y_{i\beta}^L(x)A_i(\beta, \phi)\,\theta(x)\theta(L-x),$$

$$Y_{1\beta}^R(x) = -ie^{-\beta/2}e^{-im_0 x \sinh\beta}, \quad Y_{2\beta}^R = ie^{\beta/2}e^{im_0 x \sinh\beta},$$

$$Y_{1\beta}^L(x) = e^{\beta/2}e^{-im_0 x \sinh\beta}, \quad Y_{2\beta}^L = -e^{-\beta/2}e^{im_0 x \sinh\beta}. \tag{A9}$$

Here $\theta(x)$ is the Heaviside function, with $\theta(x) = 1$ for $x > 0$ and $\theta(0) = 1/2$. $\mathcal{N}$ is a normalization constant and $A_i(\beta, \phi)$ are amplitudes which are constrained by the boundary conditions (A3), (A5) as we shall see below. We find that the ansatz (A9) satisfies (A8), provided

$$E = m_0 \cosh \beta. \tag{A10}$$

In addition, one obtains the following set of equations due to the presence of the boundaries:

$$\Psi_R^\dagger(0) \sum_i Y_{i\beta}^R(0) A_i(\beta, \phi) - \Psi_L^\dagger(0) \sum_i Y_{i\beta}^L(0) A_i(\beta, \phi) = 0, \tag{A11}$$

$$\Psi_R^\dagger(L) \sum_i Y_{i\beta}^R(L) A_i(\beta, \phi) - \Psi_L^\dagger(L) \sum_i Y_{i\beta}^L(L) A_i(\beta, \phi) = 0. \tag{A12}$$

Using the boundary conditions (A3) in the above equations, which corresponds to applying OBC, we obtain the relation between the amplitudes $A_1(\beta, \phi), A_2(\beta, \phi)$ and also the Bethe equation in the one particle sector, which are respectively given by

$$\frac{A_1(\beta, \phi)}{A_2(\beta, \phi)} = \frac{\cosh\left(\frac{1}{2}(\beta + i\phi + i\pi/2)\right)}{\cosh\left(\frac{1}{2}(\beta - i\phi - i\pi/2)\right)}, \tag{A13}$$

$$e^{2im_0 L \sinh \beta} = \frac{\cosh\left(\frac{1}{2}(\beta + i\phi + i\pi/2)\right)}{\cosh\left(\frac{1}{2}(\beta - i\phi - i\pi/2)\right)} \frac{\cosh\left(\frac{1}{2}(\beta + i\phi' + i\pi/2)\right)}{\cosh\left(\frac{1}{2}(\beta - i\phi' - i\pi/2)\right)}. \tag{A14}$$

Using the relations (A3), we have

$$e^{2im_0 L \sinh \beta} = \left(\frac{\cosh\left(\frac{1}{2}(\beta + iu)\right)}{\cosh\left(\frac{1}{2}(\beta - iu)\right)}\right)^2. \tag{A15}$$

## 2. Two particle sector

Now consider the two particle case. Due to the interactions in the Hamiltonian, the ordering of the particles is important. The wavefunction in the two particle sector can be written as

$$|2\rangle = \sum_{\alpha_1, \alpha_2 = L, R} \int_0^L \int_0^L dx_1 dx_2 \; \Psi_{\alpha_1}^\dagger(x_1) \Psi_{\alpha_2}^\dagger(x_2) \mathcal{A} \Upsilon_{\beta_1 \beta_2}^{\alpha_1 \alpha_2}(x_1, x_2) |vac\rangle, \tag{A16}$$

where $\beta_1, \beta_2$ are the rapidities and $\mathcal{A}$ is the antisymmetrizer $\mathcal{A} \Upsilon_{\beta_1 \beta_2}^{\alpha_1 \alpha_2}(x_1, x_2) = \Upsilon_{\beta_1 \beta_2}^{\alpha_1 \alpha_2}(x_1, x_2) - \Upsilon_{\beta_1 \beta_2}^{\alpha_2 \alpha_1}(x_2, x_1)$.

Applying the Hamiltonian (A1) to the two particle wavefunction (A16) we obtain the following set of equations:

$$\begin{aligned}
(-i(\partial_{x_1} + \partial_{x_2}) - (E_1 + E_2))\mathcal{A}\Upsilon_{\beta_1\beta_2}^{RR}(x_1, x_2) - im_0 \left(\mathcal{A}\Upsilon_{\beta_1\beta_2}^{RL}(x_1, x_2) + \mathcal{A}\Upsilon_{\beta_1,\beta_2}^{LR}(x_1, x_2)\right) &= 0, \\
(-i(\partial_{x_1} + \partial_{x_2}) - (E_1 + E_2))\mathcal{A}\Upsilon_{\beta_1\beta_2}^{LL}(x_1, x_2) + im_0 \left(\mathcal{A}\Upsilon_{\beta_1\beta_2}^{RL}(x_1, x_2) + \mathcal{A}\Upsilon_{\beta_1,\beta_2}^{LR}(x_1, x_2)\right) &= 0, \\
(i(\partial_{x_1} - \partial_{x_2}) - (E_1 + E_2))\mathcal{A}\Upsilon_{\beta_1\beta_2}^{LR}(x_1, x_2) + im_0 \left(-\mathcal{A}\Upsilon_{\beta_1\beta_2}^{LL}(x_1, x_2) + \mathcal{A}\Upsilon_{\beta_1,\beta_2}^{RR}(x_1, x_2)\right) & \\
+ g\delta(x_1 - x_2) \left(\mathcal{A}\Upsilon_{\beta_1\beta_2}^{LR}(x_1, x_2) - \mathcal{A}\Upsilon_{\beta_1,\beta_2}^{RL}(x_1, x_2)\right) &= 0, \\
(-i(\partial_{x_1} - \partial_{x_2}) - (E_1 + E_2))\mathcal{A}\Upsilon_{\beta_1\beta_2}^{RL}(x_1, x_2) + im_0 \left(-\mathcal{A}\Upsilon_{\beta_1\beta_2}^{LL}(x_1, x_2) + \mathcal{A}\Upsilon_{\beta_1,\beta_2}^{RR}(x_1, x_2)\right) & \\
+ g\delta(x_1 - x_2) \left(\mathcal{A}\Upsilon_{\beta_1\beta_2}^{RL}(x_1, x_2) - \mathcal{A}\Upsilon_{\beta_1,\beta_2}^{LR}(x_1, x_2)\right) &= 0.
\end{aligned} \tag{A17}$$

In order to solve the above equations, we consider the following ansatz for $\mathcal{A}\Upsilon_{\beta_1\beta_2}^{\alpha_a\alpha_2}(x_1, x_2)$:

$$\mathcal{A}\Upsilon_{\beta_1\beta_2}^{\alpha_a\alpha_2}(x_1, x_2) = \sum_{ij} \left( Y_{i\beta_1}^{\alpha_1}(x_1)Y_{j\beta_2}^{\alpha_2}(x_2) - Y_{i\beta_1}^{\alpha_2}(x_2)Y_{j\beta_2}^{\alpha_1}(x_1) \right) \left( A_{12}^{ij}\theta(x_1 - x_2) + A_{21}^{ij}\theta(x_2 - x_1) \right)$$
$$\theta(x_1)\theta(L - x_1)\theta(x_2)\theta(L - x_2),$$

(A18)

where $Y_{i\beta}^{L,R}(x)$ are given by (A9). Using the above ansatz in equations (A17) and applying the boundary conditions (A3), we obtain the Bethe equations in the two particle sector:

$$e^{2im_0 L \sinh \beta_i} = \left( \frac{\cosh\left(\frac{1}{2}(\beta_i + iu)\right)}{\cosh\left(\frac{1}{2}(\beta_i - iu)\right)} \right)^2 \frac{\sinh\left(\frac{1}{2}(\beta_i - \beta_j + 2iu)\right)}{\sinh\left(\frac{1}{2}(\beta_i - \beta_j - 2iu)\right)} \frac{\sinh\left(\frac{1}{2}(\beta_i + \beta_j + 2iu)\right)}{\sinh\left(\frac{1}{2}(\beta_i + \beta_j - 2iu)\right)},$$

(A19)

where $i, j = 1, 2$ and $u = \pi/2 - \tan^{-1}(g/2)$. The energy of the eigenstate in the two particle sector is given by $E = E_1 + E_2 = m_0 \cosh(\beta_1) + m_0 \cosh(\beta_2)$.

### 3. Bethe equations in the $N$ particle sector

Similarly, one can construct $N$ particle wavefunction

$$|N\rangle = \sum_{\alpha_1,...,\alpha_N=L,R} \int_0^L dx_1 \ldots dx_N \mathcal{A}\Upsilon_{\beta_1...\beta_N}^{\alpha_1...\alpha_N}(x_1,...x_N)\Psi_{\alpha_1}^\dagger(x_1)\ldots\Psi_{\alpha_N}^\dagger(x_N)|vac\rangle$$

$$\Upsilon_{\beta_1...\beta_N}^{\alpha_1...\alpha_N}(x_1,...x_N) = \sum_Q \theta(\{x_{Q(j)}\}) \sum_{\delta_1,...,\delta_N=1,2} A_Q^{\delta_1...\delta_N} \prod_{i=1}^N Y_{\delta_i,\beta_i}^{\alpha_i}(x_i).$$

(A20)

where $\mathcal{A}$ represents antisymmetrization with respect to the indices $\alpha_i, x_i$, while $Q$ corresponds to different orderings of particles in the configuration space. Following a similar procedure to those described above we obtain the following Bethe equations in the $N$ particle sector:

$$e^{2im_0 L \sinh \beta_i} = \left( \frac{\cosh\left(\frac{1}{2}(\beta_i + iu)\right)}{\cosh\left(\frac{1}{2}(\beta_i - iu)\right)} \right)^2 \prod_{i\neq j, j=1}^N \frac{\sinh\left(\frac{1}{2}(\beta_i - \beta_j + 2iu)\right)}{\sinh\left(\frac{1}{2}(\beta_i - \beta_j - 2iu)\right)} \frac{\sinh\left(\frac{1}{2}(\beta_i + \beta_j + 2iu)\right)}{\sinh\left(\frac{1}{2}(\beta_i + \beta_j - 2iu)\right)}.$$

(A21)

An eigenstate of the Hamiltonian corresponds to a unique set of the Bethe roots $\{\beta_j\}$, which are solutions to the Bethe equations (A21). The energy of the eigenstate is given by

$$E = \sum_{j=1}^N m_0 \cosh(\beta_j).$$

(A22)

Recall that the above Bethe equations correspond to $m_0 > 0$ and OBC (A3). Had we instead considered the case of $m_0 < 0$ with OBC (A3), or equivalently the case of $m_0 > 0$ with $\widehat{\text{OBC}}$ (A5), we would have obtained the following set of Bethe equations:

$$e^{2im_0 L \sinh \beta_i} = \left( \frac{\sinh\left(\frac{1}{2}(\beta_i + iu)\right)}{\sinh\left(\frac{1}{2}(\beta_i - iu)\right)} \right)^2 \prod_{i\neq j, j=1}^N \frac{\sinh\left(\frac{1}{2}(\beta_i - \beta_j + 2iu)\right)}{\sinh\left(\frac{1}{2}(\beta_i - \beta_j - 2iu)\right)} \frac{\sinh\left(\frac{1}{2}(\beta_i + \beta_j + 2iu)\right)}{\sinh\left(\frac{1}{2}(\beta_i + \beta_j - 2iu)\right)}.$$

(A23)

with energy of the eigenstates the same as before (A22). As we shall demonstrate, the solutions to the above Bethe equations (A21), (A23) correspond to the states with negative charge. To obtain the states with positive charge one needs to consider charge conjugated fermions. Under charge conjugation

$$\mathcal{C}\Psi^\dagger_{L,R} = \Psi'^\dagger_{L,R}(x) = \Psi_{L,R}(x), \tag{A24}$$

the Hamiltonian remains invariant. However the boundary conditions on the fermion fields $\Psi'_{L,R}$ are not the complex conjugates of (A3),(A5). Rather, we find that charge conjugation keeps the anomalous term intact, i.e. the OBC (A3) and $\widehat{\mathrm{OBC}}$ (A5) now take the form:

$$\Psi'_L(0) = e^{i\phi}\Psi'_R(0), \quad \Psi'_L(L) = -e^{-i\phi'}\Psi'_R(L),$$
$$\phi = u - \pi/2, \quad \phi' = u - \pi/2. \tag{A25}$$

and

$$\Psi'_L(0) = e^{i\phi}\Psi'_R(0), \quad \Psi'_L(L) = -e^{-i\phi'}\Psi'_R(L),$$
$$\phi = u + \pi/2, \quad \phi' = u + \pi/2 \tag{A26}$$

respectively. As mentioned in Sec. II B of the main text, with the imposition of these boundary conditions, the system is invariant under charge conjugation. Just as in the case of the original fermions $\Psi_{L,R}(x)$ considered above, one needs to construct the Bethe wavefunction for the charge conjugated fermions $\Psi'_{L,R}(x)$ with their corresponding boundary conditions given in (A25), (A26) and thus obtain the associated Bethe equations. Due to the charge conjugation invariance of the system, we find that the Bethe equations corresponding to the charge conjugated fermions are exactly same (A21), (A23) as those of the original fermions.

The charge of a state corresponding to the charge conjugated fermions can be expressed in terms of the original fermions as

$$\sum_{i=L,R} \int dx\ \Psi'^\dagger_i(x)\Psi'_i(x) = \mathrm{Const} - \sum_{i=L,R} \int dx\ \Psi^\dagger_i(x)\Psi_i(x). \tag{A27}$$

Thus, up to an unimportant constant the charge of the charge conjugated fermions $\Psi'_{L,R}(x)$ is exactly opposite to that of the original fermions $\Psi_{L,R}(x)$.

## Appendix B: Solutions to the Bethe equations in the trivial phase

As mentioned above, the Bethe equations in the trivial phase are given by (A23). Each eigenstate of the Hamiltonian corresponds to a unique set of roots $\{\beta_j\}$ called Bethe roots, which satisfy the Bethe equations (A21), (A23). Due to the presence of open boundary conditions, the Bethe equations are reflection symmetric: $\beta_i \leftrightarrow -\beta_i$. As a result, the solutions to the Bethe equations (A21), (A23) come in pairs $(\beta_i, -\beta_i)$. In the thermodynamic limit $N, L \to \infty$, the Bethe roots form a dense set and the Bethe equations can be expressed as integral equations, which can be solved by Fourier transform. As we shall see, in this limit one also needs to introduce a cutoff $\Lambda = \pi N/L \gg m_0$ on $\beta_j$. From the relation between the Bethe roots $\beta_j$ and the energy (A22), we can infer that the roots lying on the line $\beta_j = \alpha_j + i\pi$ have lower energy. Although this statement is in general true, one needs to be careful in accounting for the boundary bound states which are described by purely imaginary solutions to the Bethe equations called boundary strings. In this work we present our solution for the OBC (A3) in two regimes of the bare mass parameter $m_0 > 0$ and $m_0 < 0$. Due to the duality (A4), these two cases correspond to $m_0 < 0$ and $m_0 > 0$ respectively with $\widehat{\mathrm{OBC}}$ (A5).

When $m_0 < 0$ with OBC, which is described by the Bethe equations (A23), the ground state is non-degenerate and there exist no boundary bound states. We refer to this phase as the *trivial phase*. In contrast, when $\widehat{\mathrm{OBC}}$ are applied, which is described by the Bethe equations (A21), the system hosts exponentially localized zero energy modes (ZEMs) at each edge. In the entire repulsive regime ($u < \pi/2$) and in the attractive regime for $\pi/2 < u < 2\pi/3$, these ZEMs are related to the existence of what we call 'close or short boundary string' ([36],[37]) solutions of the Bethe equations. In the attractive regime for $u > 2\pi/3$, in addition to the close boundary strings, the ZEMs are associated with a new type of boundary strings named 'breather boundary strings'. As a result, there exists a fourfold degenerate ground state. We refer to this phase as the *topological phase*. The situation for the $m_0 > 0$ case is reversed and can be obtained with use of the duality transformation.

Let us consider the Bethe equations corresponding to the charge conjugated fermions for the case $m_0 < 0$ with OBC, which are given by (A23). As mentioned before, from the relation between the Bethe roots $\beta_j$ and the energy (A22), we can infer that the roots lying on the line $\beta_j = \alpha_j + i\pi$ have lower energy. Since we are interested in the low energy spectrum, we consider the state which has all the roots lying on this line. By making the transformation $\beta_j \to \alpha_j + i\pi$ and taking a logarithm, we obtain the logarithmic form of the Bethe equations

$$-2im_0L\sinh\alpha_i + 2i\pi n_i = 2\ln\left(\frac{\cosh\left(\frac{1}{2}(\alpha_i + iu)\right)}{\cosh\left(\frac{1}{2}(\alpha_i - iu)\right)}\right) + \ln\left(\frac{\sinh\left(\alpha_i - iu\right)}{\sinh\left(\alpha_i + iu\right)}\right) + \sum_{\sigma=\pm}\sum_j \ln\left(\frac{\sinh\left(\frac{1}{2}(\alpha_i - \sigma\alpha_j + 2iu)\right)}{\sinh\left(\frac{1}{2}(\alpha_i - \sigma\alpha_j - 2iu)\right)}\right).$$

where $n_i$ are integers corresponding to the logarithmic branch. These integers are in one-to-one correspondence with the Bethe roots $\alpha_i$. Note that the selection rule $i \neq j$, which is necessary for a non-vanishing wavefunction, has been lifted by adding a counter term that cancels the term arising from $\alpha_i = \alpha_j$. The trivial solution $\alpha_i = 0$ to (A23) leads to a vanishing wave function [33]. Hence, this solution needs to be removed, and this can be achieved by omitting the integer $n_i$ corresponding to $\alpha_i = 0$. As we shall see, this leads to a delta function centered at $\alpha = 0$ in the density distribution of roots to be defined below.

Using the notation $\varphi(x, y) = i\ln\left(\frac{(\sinh(x+iy)/2)}{(\sinh(x-iy)/2)}\right)$, the above equation can be written as

$$2m_0L\sinh\alpha_i = 2\varphi(\alpha_i, u+\pi) + \sum_{\sigma=\pm}\sum_j \varphi(\alpha_i - i + \sigma\alpha_j, 2u) - \varphi(2\alpha_i, 2u) + 2\pi n_i. \tag{B1}$$

Subtracting the equation (B1) for the root $\alpha_{i+1}$ from that of the root $\alpha_i$, we have

$$2m_0L\left(\sinh\alpha_{i+1} - \sinh\alpha_i\right) = 2\varphi\left(\alpha_{i+1}, u+\pi\right) - 2\varphi\left(\alpha_i, u+\pi\right)$$
$$-(\varphi(2\alpha_{i+1}, 2u) - \varphi(2\alpha_i, 2u)) + \sum_{\sigma=\pm}\sum_j(\varphi(\alpha_{i+1} + \sigma\alpha_j, 2u) - \varphi(\alpha_i + \sigma\alpha_j, 2u)) + 2\pi(n_{i+1} - n_i). \tag{B2}$$

In this state the integers $n_i$ are consecutively filled without any gap except for the integer corresponding to $\alpha_i = 0$ as mentioned above. In the thermodynamic limit as mentioned above, Bethe roots form a dense set. In this limit one can define the density distribution of roots as

$$\rho(\alpha) = \frac{1}{2L}\frac{1}{(\alpha_{i+1} - \alpha_i)}, \tag{B3}$$

where the factor of 2 is to account for the doubling of the solutions mentioned above. Using this in the equation (B2), we have

$$2m_0\cosh\alpha - \frac{1}{L}\varphi'(\alpha, u+\pi) - 4\pi\rho_{|0\rangle}(\alpha) - \frac{1}{L}\delta(\alpha) + \frac{1}{L}\varphi'(\alpha, u)$$
$$= \sum_\sigma\sum_j \frac{1}{L}\varphi'(\alpha + \sigma\gamma_j, 2u) = \sum_\sigma \int \varphi'(\alpha + \sigma\gamma, 2u)\rho_{|0\rangle}(\gamma)d\gamma, \tag{B4}$$

where

$$\varphi'(x, y) = \partial_x\varphi(x, y) = \frac{\sin y}{\cosh x - \cos y}, \quad \varphi'(x, y+\pi) = \partial_x\varphi(x, y+\pi) = -\frac{\sin y}{\cosh x + \cos y}. \tag{B5}$$

Note that we labelled the density distribution by $\rho_{|0\rangle}(\alpha)$. The reason for this labelling will be discussed shortly. The above equation (B4) can be solved by applying a Fourier transform. The Fourier transforms of all the terms in the above equation are well defined except for the the $\cosh(\alpha)$ term. In order to tackle this, we need to apply a cutoff on the 'rapidity' values $\alpha$ when taking the Fourier transform of this term [32]. We obtain

$$\tilde\rho_{|0\rangle}(\omega) = \frac{2m_0\tilde c(\omega) - \frac{1}{L}\tilde\varphi'(\omega, u+\pi) + \frac{1}{L}\tilde\varphi'(\omega, u) - 1}{2(2\pi + \tilde\varphi'(\omega, 2u))}, \tag{B6}$$

where

$$\tilde{c}(\omega) = \int_{-\Lambda}^{\Lambda} d\alpha \ e^{i\alpha\omega} \cosh(\alpha) = \frac{e^{\Lambda}}{2} \left( \frac{e^{i\Lambda\omega}}{1+i\omega} + \frac{e^{-i\Lambda\omega}}{1-i\omega} \right), \tag{B7}$$

$$\tilde{\varphi}'(\omega, y) = \int_{-\infty}^{\infty} dx \ e^{ix\omega} \varphi'(x, y) = 2\pi \frac{\sinh((\pi - y)\omega)}{\sinh(\pi\omega)}, \ (y < 2\pi), \tag{B8}$$

$$\tilde{\varphi}'(\omega, y + \pi) = \int_{-\infty}^{\infty} dx \ e^{ix\omega} \varphi'(x, y + \pi) = -2\pi \frac{\sinh(y\omega)}{\sinh(\pi\omega)}, \ (y < \pi). \tag{B9}$$

Explicitly, we have

$$\tilde{\rho}_{|0\rangle}(\omega) = \tilde{\rho}_{Bulk}(\omega) + \tilde{\rho}_{Boundary}^{tr}(\omega), \tag{B10}$$

where

$$\tilde{\rho}_{Bulk}(\omega) = \frac{m_0}{4\pi} \frac{\sinh(\pi\omega)}{\sinh(\pi\omega) + \sinh((\pi - 2u)\omega)} e^{\Lambda} \left( \frac{e^{i\Lambda\omega}}{1+i\omega} + \frac{e^{-i\Lambda\omega}}{1-i\omega} \right), \tag{B11}$$

$$\tilde{\rho}_{Boundary}^{tr}(\omega) = \frac{1}{2L} \frac{\sinh((\pi - u)\omega) + \sinh(u\omega) - \sinh(\pi\omega)}{\sinh(\pi\omega) + \sinh((\pi - 2u)\omega)}, \tag{B12}$$

The charge of the state corresponding to the charge conjugated fermions obtained above by filling the roots continuously on the $i\pi$ line, which is described by the density distribution $\rho_{|0\rangle}(\alpha)$ is

$$L \int_{-\Lambda}^{\Lambda} d\alpha \ \rho_{|0\rangle}(\alpha) = L\tilde{\rho}_{|0\rangle}(0) = q\frac{m_0 L e^{\Lambda}}{2\pi}, \ q = \frac{\pi}{2(\pi - u)}. \tag{B13}$$

The total charge is divergent in the limit $\Lambda \to \infty$, and hence normal ordering is needed. We define the normal ordered charge $\mathcal{N}$ (in units of the renormalized charge of the physical excitations $q$) of the state as

$$\mathcal{N} = -\frac{1}{q} \left\{ \tilde{\rho}_{|0\rangle}(0) - \tilde{\rho}_{Bulk}(0) \right\}. \tag{B14}$$

With this definition, the normal ordered charge of the state obtained above is given by:

$$\mathcal{N} = 0. \tag{B15}$$

This allows us to label this state as $|0\rangle$. This is the unique ground state exhibited by the system. Hence we find that the above described solution to the Bethe equations corresponding to the trivial phase (A23) gives a state which we labelled $|0\rangle$.

### 1. Solitons and the mass gap

The simplest excitations in the bulk constitute of solitons. A soliton can be created on top of a state by removing a root from the density distribution corresponding to that particular state. This is also referred to as adding a hole. Let us remove a root from the density distribution corresponding to the state $|0\rangle$ described above. Removing a root is equivalent to omitting an integer $n_i$ in (B1) corresponding to that particular root. As mentioned before, an omitted integer gives rise to a delta function in the density distribution. Consider a state with one hole at rapidity $\alpha = \theta$ (or equivalently removing the root $\theta$). Denoting the density distribution of the resulting state by $\rho_\theta(\alpha)$, we have

$$2m_0 \cosh\alpha - \frac{1}{L}\varphi'(\alpha, u + \pi) - 4\pi\rho_\theta(\alpha) - \frac{1}{L}\delta(\alpha) + \frac{1}{L}\varphi'(\alpha, u)$$
$$-\frac{1}{L}\sum_{\sigma=\pm} \delta(\alpha + \sigma\theta) = \sum_\sigma \int \varphi'(\alpha + \sigma\gamma, 2u)\rho_\theta(\gamma)d\gamma. \tag{B16}$$

This equation can be solved by following the same procedure described above. We obtain

$$\tilde{\rho}_\theta(\omega) = \tilde{\rho}_{|0\rangle}(\omega) + \Delta\tilde{\rho}_\theta, \quad \Delta\tilde{\rho}_\theta(\omega) = -\frac{1}{2L}\frac{\sinh(\pi\omega)\cos(\theta\omega)}{\sinh((\pi-u)\omega)\cosh(u\omega)}, \tag{B17}$$

where $\tilde{\rho}_{|0\rangle}(\omega)$ is given by (B10). We see that by adding a soliton (hole) with rapidity $\alpha = \theta$ to the state $|0\rangle$, the associated density distribution $\rho_{|0\rangle}(\alpha)$ undergoes a change $\Delta\rho_\theta(\alpha)$, resulting in a new density distribution $\rho_\theta(\alpha)$. The charge associated with the state described by this distribution is

$$L\tilde{\rho}_\theta(0) = L\tilde{\rho}_{|0\rangle}(0) + L\Delta\tilde{\rho}_\theta(0) = L\tilde{\rho}_{|0\rangle}(0) - \frac{\pi}{2(\pi-u)}. \tag{B18}$$

Using the expression for the charge associated with the state $|0\rangle$ (B10) in the equation (B18), we find that the charge associated with the state containing the soliton is

$$L\tilde{\rho}_\theta(0) = q\left(\frac{m_0 L e^\Lambda}{2\pi} - 1\right), \quad q = \frac{\pi}{2(\pi-u)}. \tag{B19}$$

Defining the normal ordered charge as before, we find that the normal ordered charge of the state containing a soliton on top of the state $|0\rangle$ is

$$\mathcal{N} = 1. \tag{B20}$$

One defines the charge of a soliton as the change in the charge of a state due to the addition of a hole. Comparing the charges of the state with and without the hole, i.e., (B19) and (B13) respectively, we find that the charge of a soliton is

$$|L\Delta\tilde{\rho}_\theta(0)| = \frac{\pi}{2(\pi-u)} = q. \tag{B21}$$

The energy of the soliton is given by

$$E_\theta = -Lm_0\int_{-\Lambda}^{\Lambda} d\alpha \ \cosh\alpha \ \Delta\rho_\theta(\alpha). \tag{B22}$$

This can be expressed in terms of the Fourier transforms of $\cosh\alpha$ and $\Delta\rho_\theta(\alpha)$ as

$$E_\theta = \frac{m_0}{4\pi}\int_{-\infty}^{\infty} d\omega \ \frac{e^\Lambda}{2}\left(\frac{e^{i\Lambda\omega}}{1+i\omega} + \frac{e^{-i\Lambda\omega}}{1-i\omega}\right)\frac{\sinh(\pi\omega)\cos(\theta\omega)}{\sinh((\pi-u)\omega)\cosh(u\omega)}, \tag{B23}$$

where (B9) and (B17) were used. In the limit $\Lambda \to \infty$, the poles close to the $x-$axis arising from the term $\cosh(u\omega)$ contribute, whereas the contribution from the other terms vanishes exponentially [32]. The integration is performed by closing the contour in the upper and lower half planes for the first and the second terms of $\tilde{c}(\omega)$ respectively. We obtain

$$E_\theta = m\cosh(\gamma\theta), \quad m = \frac{m_0\gamma}{\pi(\gamma-1)}\tan(\pi\gamma) \ e^{\Lambda(1-\gamma)}, \quad \gamma = \frac{\pi}{2u}, \tag{B24}$$

where $m$ is the renormalized mass. Instead of considering the Bethe equations corresponding to charge conjugated fermions, by considering the Bethe equations of the original fermions and adding a hole one obtains anti-solitons which carry charge $-q$, which is exactly opposite to that of the solitons. The energy of the anti-solitons is exactly same as that of the solitons (B24). Hence we see that the first excited state above the ground state $|0\rangle$ is doubly degenerate with normal ordered charge $\mathcal{N} = \pm 1$, where $\pm$ correspond to the soliton and anti-soliton respectively.

## Appendix C: Solutions to the Bethe equation in the topological phase

The Bethe equations in the topological phase are given by (A21). The solution to these equations is described in detail below.

### 1. The states $|\pm 1\rangle$

Let us consider the Bethe equations corresponding to the charge conjugated fermions (A21). As mentioned before, from the relation between the Bethe roots $\beta_j$ and the energy (A22), we can infer that the roots lying on the line $\beta_j = \alpha_j + i\pi$ have lower energy. Since we are interested in the low energy spectrum, we consider the state which has all the roots lying on this line. By making the transformation $\beta_j \to \alpha_j + i\pi$ and applying logarithm to (A21), we obtain the logarithmic form of the Bethe equations in this phase

$$-2im_0 L \sinh \alpha_i = \ 2 \ln \left( \frac{\sinh\left(\frac{1}{2}(\alpha_i+iu)\right)}{\sinh\left(\frac{1}{2}(\alpha_i-iu)\right)} \right) - 2i\pi n_i + \ln \left( \frac{\sinh(\alpha_i-iu)}{\sinh(\alpha_i+iu)} \right) - \sum_{\sigma=\pm} \sum_j \ln \left( \frac{\sinh\left(\frac{1}{2}(\alpha_i-\sigma\alpha_j-2iu)\right)}{\sinh\left(\frac{1}{2}(\alpha_i-\sigma\alpha_j+2iu)\right)} \right). \quad \text{(C1)}$$

Using the same notation as before, $\varphi(x,y) = i\ln\left(\frac{(\sinh(x+iy)/2)}{(\sinh(x-iy)/2)}\right)$, the above equation can be written as

$$2m_0 L \sinh \alpha_i = 2\varphi(\alpha_i,u) + \sum_{\sigma=\pm} \sum_j \varphi(\alpha_i - i + \sigma\alpha_j, 2u) - \varphi(2\alpha_i, 2u) + 2\pi n_i. \quad \text{(C2)}$$

The above equation can be solved by following the same procedure as in the trivial phase. Let us denote the density distribution of the resulting state by $\rho_{|1\rangle}(\alpha)$. The reason for this notation will become evident soon. We obtain

$$2m_0 \cosh \alpha - \frac{2}{L}\varphi'(\alpha,u) - 4\pi\rho_{|1\rangle}(\alpha) - \frac{1}{L}\delta(\alpha) + \frac{1}{L}\varphi'(\alpha,u) + \frac{1}{L}\varphi'(\alpha, u+\pi)$$
$$= \sum_\sigma \sum_j \frac{1}{L}\varphi'(\alpha + \sigma\gamma_j, 2u) = \sum_\sigma \int \varphi'(\alpha + \sigma\gamma, 2u)\rho_{|1\rangle}(\gamma)d\gamma, \quad \text{(C3)}$$

where

$$\varphi'(x,y) = \partial_x \varphi(x,y) = \frac{\sin y}{\cosh x - \cos y}, \quad \varphi'(x,y+\pi) = \partial_x \varphi(x, y+\pi) = -\frac{\sin y}{\cosh x + \cos y}. \quad \text{(C4)}$$

The above equation (C3) can be solved by applying a Fourier transform just as in the previous case. Following the same procedure, we obtain

$$\tilde{\rho}_{|1\rangle}(\omega) = \tilde{\rho}_{Bulk}(\omega) + \tilde{\rho}^{top}_{Boundary}(\omega), \quad \text{(C5)}$$

where $\tilde{\rho}_{Bulk}(\omega)$ is same as in the trivial phase (B11), whereas

$$\tilde{\rho}^{top}_{Boundary}(\omega) = \frac{1}{2L} \frac{-\sinh((\pi-u)\omega) - \sinh(u\omega) - \sinh(\pi\omega)}{\sinh(\pi\omega) + \sinh((\pi - 2u)\omega)}, \quad \text{(C6)}$$

The charge of the state obtained above by filling the roots continuously on the $i\pi$ line, which is described by the density distribution $\rho_{|1\rangle}(\alpha)$ is

$$L \int_{-\Lambda}^{\Lambda} d\alpha \ \rho_{|1\rangle}(\alpha) = L\tilde{\rho}_{|1\rangle}(0) = \frac{m_0 L e^\Lambda}{2\pi} \frac{\pi}{2(\pi - u)} - \frac{\pi}{2(\pi - u)}. \quad \text{(C7)}$$

The first term which is divergent in the limit $\Lambda \to \infty$ is the charge associated with the bulk. The second term is due to the charge associated with the boundaries. This can be expressed as

$$L\tilde{\rho}_{|1\rangle}(0) = q\left(\frac{m_0 L e^\Lambda}{2\pi} - 1\right), \quad q = \frac{\pi}{2(\pi - u)}. \quad \text{(C8)}$$

Defining the normal ordered charge of the state as before, we have

$$\mathcal{N} = 1. \quad \text{(C9)}$$

This allows us to label this state as $|1\rangle$. Due to charge conjugation symmetry, there exists another degenerate state with total normal ordered charge $\mathcal{N} = -1$. This state can be obtained by starting with the Bethe equations corresponding to the original fermions, instead of charge conjugated fermions considered above, and following the same procedure as above.

## 2. The states $|0\rangle_+$ and $|0\rangle_-$

On top of the two states $|\pm 1\rangle$ obtained above, there exist two states $|0\rangle_\pm$ with total normal ordered charge $\mathcal{N} = 0$, that are degenerate with the states $|\pm 1\rangle$. The construction of these two states differs for different ranges of the values of $u$ which are given by

$$\left(\frac{2n}{2n+1}\right)\pi < u < \left(\frac{2n+2}{2n+3}\right)\pi, \ n \in \mathbb{N}. \tag{C10}$$

Here $\mathbb{N}$ represents all natural numbers including zero. Below we provide the explicit construction of the two states $|0\rangle_\pm$ for all values of $n$.

### a. $0 < u < 2\pi/3$

Let us first consider the case $n = 0$ in (C10). The Bethe equations (A21) exhibit one boundary string solution $\lambda_{(0)} = \pm iu$, which corresponds to a double pole. (For boundary strings we use the notation $\lambda = \alpha + 2im\pi$, where $m \in \mathbb{Z}$.) In the range $0 < u < 2\pi/3$, this boundary string falls under the *close or short boundary string* category [36, 37]. Essentially, the boundary string whose total charge is equal and opposite to that of a soliton is categorized as a "close boundary string." The close boundary string corresponds to a bound state which is exponentially localized at both the edges in a symmetric superposition. Since it corresponds to a double pole, there exists another state which is the anti-symmetric superposition of the bound state localized at both the edges [24]. Note that although the Bethe equations, and hence the boundary strings are valid throughout this regime $0 < u < 2\pi/3$, due to the cutoff procedure being used, we are still restricted to the values of $u > \pi/3$ in the current regularization scheme. Consider adding the boundary string $\lambda_{(0)}$ to the state described by the density distribution $\rho_{|1\rangle}(\alpha)$. Labelling the density distribution associated with the resulting state as $\rho_{CBS0}(\alpha)$, we have

$$2m_0 \cosh\alpha - \frac{1}{L}\varphi'(\alpha, u) - 4\pi\rho_{CBS0}(\alpha) - \frac{1}{L}\delta(\alpha) + \frac{1}{L}\varphi'(\alpha, u+\pi)$$
$$= \frac{1}{L}\varphi'(\alpha, 3u) + \frac{1}{L}\varphi'(\alpha, u) + \sum_\sigma \int \varphi'(\alpha + \sigma\gamma, 2u)\rho_{CBS0}(\gamma)d\gamma. \tag{C11}$$

Following the same procedure as above, we obtain

$$\tilde{\rho}_{CBS0}(\omega) = \tilde{\rho}_{|1\rangle}(\omega) + \Delta\tilde{\rho}_{CBS0}(\omega), \quad \Delta\tilde{\rho}_{CBS0}(\omega) = -\frac{1}{L}\frac{\sinh((\pi - 2u)\omega)\cosh(u\omega)}{\sinh(\pi\omega) + \sinh((\pi - 2u)\omega)}, \tag{C12}$$

where $\tilde{\rho}_{|1\rangle}(\omega)$ is given by (C5), (C6). We see that adding the boundary string $\lambda_{(0)} = \pm iu$ to the state $|1\rangle$ results in the change $\Delta\rho_{CBS0}(\alpha)$ to the root distribution $\rho_{|1\rangle}(\alpha)$ due to the back flow effect. This results in a new density distribution $\rho_{CBS0}(\alpha)$ for the Bethe roots lying on the $i\pi$ line. Hence, we obtain a new state which is described by the set of Bethe roots which includes the roots on the $i\pi$ line with the density distribution $\rho_{CBS0}(\alpha)$ and the boundary string $\lambda_{(0)} = \pm iu$. The total charge of this state is

$$1 + L\tilde{\rho}_{CBS0}(0) = \frac{m_0 Le^\Lambda}{2\pi}\frac{\pi}{2(\pi - u)}. \tag{C13}$$

The first term '1' on the left side of the above equation corresponds to the boundary string and the second term $L\tilde{\rho}_{CBS0}(0)$ corresponds to the roots on the $i\pi$ line. Hence the normal ordered charge of this state is

$$\mathcal{N} = 0. \tag{C14}$$

Comparing the charge of this state with the charge of the state described by the density distribution $\rho_{|1\rangle}(\alpha)$, we find that the net charge of the boundary string $\lambda_{(0)} = \pm iu$ is exactly equal and opposite to that of a soliton, and hence it is a close boundary string as mentioned above. Note that we have labelled this boundary string 'CBS0', where CBS stands for close boundary string and '0' refers to the fact that it is the fundamental boundary string (zero$^{th}$ order string)

To obtain the energy of the state described by the root distribution $\rho_{CBS0}(\alpha)$ obtained above, we need to calculate the energy of the boundary string $\lambda_{(0)} = \pm iu$. This is given by

$$E_{CBS0} = -m_0 \cosh(iu) - m_0 L \int_{-\Lambda}^{\Lambda} d\alpha \, \cosh\alpha \, \Delta\rho_{CBS0}(\alpha). \tag{C15}$$

The first term in the above expression is the bare energy, and the second term is the contribution due to the back flow effect. This can be written as

$$E_{CBS0} = -m_0 L \int_{-\Lambda}^{\Lambda} d\alpha \, \cosh\alpha \, \left( \frac{1}{2L} \sum_{\sigma=\pm} \delta(\alpha + \sigma iu) + \Delta\rho_{CBS0}(\alpha) \right). \tag{C16}$$

This can be expressed in terms of the Fourier transforms of $\cosh\alpha$ and the term in the parentheses. We have

$$E_{CBS0} = -\frac{m_0}{4\pi} \int_{-\infty}^{\infty} d\omega \, \frac{e^{\Lambda}}{2} \left( \frac{e^{i\Lambda\omega}}{1+i\omega} + \frac{e^{-i\Lambda\omega}}{1-i\omega} \right) \frac{\sinh(\pi\omega)\cosh(u\omega)}{\sinh((\pi-u)\omega)\cosh(u\omega)}. \tag{C17}$$

Comparing this equation with (B23), we see that it can be evaluated in the same way. We obtain

$$E_{CBS0} = 0. \tag{C18}$$

We see that by adding the boundary string $\lambda_{(0)} = \pm iu$ to the state $|1\rangle$, we obtain a degenerate state with total normal ordered charge $\mathcal{N} = 0$. We label this state by $|0\rangle_+$, where $+$ corresponds to the symmetric superposition of the bound state localized at both the edges. As mentioned before, there exists another degenerate state which corresponds to the anti-symmetric superposition of the bound state localized at both the edges. We label this state $|0\rangle_-$. These two states $|0\rangle_\pm$ can also be obtained by adding the boundary string $\lambda_{(0)} = \pm iu$ to the state $|-1\rangle$.

### b. $2\pi/3 < u < 4\pi/5$

Now consider the case $n = 1$ in (C10). The boundary string $\lambda_{(0)} = \pm iu$ is no longer a close boundary string when $u > 2\pi/3$, where it falls under a new category which we name "*breather boundary string*". The reason for this change in the category and the associated name will become evident soon. Note that when $u > 2\pi/3$, the Fourier transform of the first term on the right hand side of equation (C11) takes a different form compared to (B9). It is given by

$$\tilde{\varphi}'(\omega, y) = \int_{-\infty}^{\infty} dx \, e^{ix\omega} \varphi'(x, y) = 2\pi \frac{\sinh((3\pi - y)\omega)}{\sinh(\pi\omega)}, \quad (y < 4\pi). \tag{C19}$$

Hence, adding the boundary string $\lambda_{(0)} = \pm iu$, instead of (C12), we obtain the following distribution for the roots on the $i\pi$ line:

$$\tilde{\rho}_{BBS0}(\omega) = \tilde{\rho}_{|1\rangle}(\omega) + \Delta\tilde{\rho}_{BBS0}(\omega), \quad \Delta\tilde{\rho}_{BBS0}(\omega) = -\frac{1}{L} \frac{\sinh(2(\pi-u)\omega)\cosh((\pi-u)\omega)}{\sinh(\pi\omega) + \sinh((\pi-2u)\omega)}. \tag{C20}$$

The total charge of this state is

$$1 + L\tilde{\rho}_{BBS0}(0) = L\tilde{\rho}_{|1\rangle}(0) = \frac{m_0 L e^{\Lambda}}{2\pi} \frac{\pi}{2(\pi-u)} - \frac{\pi}{2(\pi-u)}. \tag{C21}$$

The total normal ordered charge of the state is

$$\mathcal{N} = 1. \tag{C22}$$

We label this state which is obtained by adding the boundary string $\lambda_{(0)}$ for $u > 2\pi/3$ as

$$|1\rangle_{+B0} . \tag{C23}$$

We see that the total charge associated with the boundary string $\lambda_{(0)} = \pm iu$ is zero when $u > 2\pi/3$. Recall that it had charge $q$ in the previous case where $u < 2\pi/3$. This change in the charge associated with the boundary string is the reason why it is no longer a close boundary string. Hence we have labelled it 'BBS0', where BBS stands for breather boundary string, and '0' refers to the fact that it is the zero$^{th}$ order string. The wavefunction associated with the boundary breather is localized at both the boundaries in a symmetric superposition, which is represented by $+$ in the subscript. Due to the double pole associated with the boundary string, there exists another degenerate state corresponding to the anti-symmetric superposition represented by $|1\rangle_{-B0}$. Due to charge conjugation symmetry, there exists degenerate states whose total charge is exactly opposite to that of the states $|1\rangle_{\pm B0}$. We label these states as $|-1\rangle_{\pm B0}$.

Now let us calculate the energy associated with this boundary string for $u > 2\pi/3$. It is given by

$$E_{BBS0} = -m_0 \cosh(iu) - m_0 L \int_{-\Lambda}^{\Lambda} d\alpha \ \cosh\alpha \ \Delta\rho_{BBS0}(\alpha), \tag{C24}$$

where again, the first term is the bare energy and the second term is the contribution due to the back flow effect. This can be written as

$$E_{BBS0} = m_0 L \int_{-\Lambda}^{\Lambda} d\alpha \ \cosh\alpha \ \left( \frac{1}{2L} \sum_{\sigma=\pm} \delta(\alpha + \sigma i(\pi - u)) - \Delta\rho_{BBS0}(\alpha) \right). \tag{C25}$$

This can be expressed in terms of the Fourier transforms of $\cosh\alpha$ and term in the parenthesis. We have

$$E_{BBS0} = -\frac{m_0}{2\pi} \int_{-\infty}^{\infty} d\omega \ \frac{e^\Lambda}{2} \left( \frac{e^{i\Lambda\omega}}{1 + i\omega} + \frac{e^{-i\Lambda\omega}}{1 - i\omega} \right) \frac{\cosh((\pi - u)\omega)\cosh(\pi\omega/2)\cosh((\pi/2 - u)\omega)}{\cosh(u\omega)}. \tag{C26}$$

This can be evaluated in the same way as before. We obtain

$$E_{BBS0} = m \sin(\pi\xi), \ \ \xi = 2\gamma - 1. \tag{C27}$$

We find that the energy of the boundary string $\lambda_{(0)} = \pm iu$ is no longer zero, and it is rather given by (C27), which is positive for $u > 2\pi/3$. We find that a new scale emerges at the boundary, which is asymptotically equal to the mass of the first bulk breather in the semi-classical limit $u \to \pi$. For $u > 2\pi/3$, since both the charge and the energy scale associated with the boundary string $\lambda_{(0)} = \pm iu$ change such that they are now equal to those corresponding to the bulk breather, we conclude that the boundary string $\lambda_{(0)}$ falls under a new category for $u > 2\pi/3$, which we name breather boundary string as mentioned before.

We observe that upon choosing one of the roots of the Bethe equations (A21) to be the boundary string $\lambda_{(0)} = \pm iu$, a new boundary string $\lambda_{(1)} = \pm 3iu$ solution emerges for $u > 2\pi/3$. Note that this "higher order boundary string" is not a solution to the Bethe equations (A21) for $u < 2\pi/3$. Adding this boundary string to the state $|1\rangle_{B0}$ obtained above, and denoting the density distribution associated with the resulting state by $\rho_{CBS1}(\alpha)$, we have

$$2m_0 \cosh\alpha - \frac{1}{L}\varphi'(\alpha, u) - 4\pi\rho_{CBS1}(\alpha) - \frac{1}{L}\delta(\alpha) + \frac{1}{L}\varphi'(\alpha, u + \pi) - \frac{1}{L}\varphi'(\alpha, 3u) - \frac{1}{L}\varphi'(\alpha, u)$$
$$= \frac{1}{L}\varphi'(\alpha, 5u) - \frac{1}{L}\varphi'(\alpha, u) + \sum_\sigma \int \varphi'(\alpha + \sigma\gamma, 2u)\rho_{CBS1}(\gamma)d\gamma. \tag{C28}$$

For $u < 4\pi/5$, the Fourier transform of the first term on the right side of the above equation is given by (C40). Following the same procedure as above, we obtain

$$\tilde{\rho}_{CBS1}(\omega) = \tilde{\rho}_{BBS0}(\omega) + \Delta\tilde{\rho}_{CBS1}(\omega), \quad \Delta\tilde{\rho}_{CBS1}(\omega) = -\frac{1}{L}\frac{\sinh((\pi - 2u)\omega)\cosh((2\pi - 3u)\omega)}{\sinh(\pi\omega) + \sinh((\pi - 2u)\omega)}, \tag{C29}$$

where $\tilde{\rho}_{BBS0}$ is given by (C20). We have obtained a new state which is described by the set of Bethe roots which in addition to roots on the $i\pi$ line with the density distribution $\rho_{CBS1}(\alpha)$, includes the boundary strings $\lambda_{(0)} = \pm iu$ and $\lambda_{(1)} = \pm 3iu$. The total charge of this state is

$$2 + L\tilde{\rho}_{CBS1}(0) = \frac{m_0 L e^{\Lambda}}{2\pi} \frac{\pi}{2(\pi - u)}. \tag{C30}$$

Here '2' on the left hand hand side corresponds to the two boundary strings $\lambda_{(0)}$ and $\lambda_{(1)}$. The normal ordered charge of this state is

$$\mathcal{N} = 0. \tag{C31}$$

Hence, we find that by adding the boundary string $\lambda_{(1)} = \pm 3iu$ to the state $|1\rangle_{+B0}$, we have obtained a state whose total charge is zero. Therefore, we find that the boundary string $\lambda_{(1)}$ is a close boundary string, and hence it is denoted by 'CBS1', where '1' refers to the fact that it is the first higher order string.

Let us now calculate the energy of the boundary string $\lambda_{(1)}$ in this regime where $2\pi/3 < u < 4\pi/5$. It is given by

$$E_{CBS1} = -m_0 \cosh(3iu) - m_0 L \int_{-\Lambda}^{\Lambda} d\alpha \ \cosh\alpha \ \Delta\rho_{CBS1}(\alpha), \tag{C32}$$

where again the first term is the bare energy and the second term is due to the back flow effect. This can be written as

$$E_{CBS1} = m_0 L \int_{-\Lambda}^{\Lambda} d\alpha \ \cosh\alpha \ \left( \frac{1}{2L} \sum_{\sigma = \pm} \delta(\alpha + \sigma i(2\pi - 3u)) - \Delta\rho_{CBS1}(\alpha) \right). \tag{C33}$$

This can be expressed in terms of the Fourier transforms of $\cosh\alpha$ and term in the parenthesis. We have

$$E_{CBS1} = -\frac{m_0}{4\pi} \int_{-\infty}^{\infty} d\omega \ \frac{e^{\Lambda}}{2} \left( \frac{e^{i\Lambda\omega}}{1 + i\omega} + \frac{e^{-i\Lambda\omega}}{1 - i\omega} \right) \frac{\cosh((2\pi - 3u)\omega)\sinh(\pi\omega)}{\sinh((\pi - u)\omega)\cosh(u\omega)}. \tag{C34}$$

This can be evaluated in the same way as before. We obtain

$$E_{CBS1} = -m \sin(\pi\xi). \tag{C35}$$

Comparing this with the energy of the breather boundary string $\lambda_{(0)}$ (C27), we see that they are exactly equal and opposite. Hence, we find that by adding the first higher order boundary string $\lambda_{(1)}$ to the state $|1\rangle_{+B0}$, which already contains the breather boundary string $\lambda_{(0)}$, we have obtained a state whose total normal ordered charge is zero and is degenerate with the states $|\pm 1\rangle$. This is precisely the state $|0\rangle_{+}$. Just as in the previous regime $u < 2\pi/3$, there exists another state $|0\rangle_{-}$ due to the double pole associated with $\lambda_{(0)}$.

### c. $(2n/2n + 1)\pi < u < (2n + 2/2n + 3)\pi, \ n \in \mathbb{N}$

Now let us consider the general case where $n$ is arbitrary. By considering the boundary string $\lambda_{(0)} = \pm iu$ solution as one of the roots, we see that a new higher order boundary string solution $\lambda_{(1)} = \pm 3iu$ arises. Considering $\lambda_{(1)}$ as one of the roots, we find that a new higher order boundary string $\lambda_{(2)} = \pm 5iu$ solution arises. The emergence of new higher order boundary string solutions occurs until we reach the boundary string of the highest order $\lambda_{(n)} = \pm i(2n + 1)u$ corresponding to the current regime. In total there exist $n + 1$ boundary strings which are given by

$$\lambda_{(j-1)} = \pm i(2j - 1)u, \ \ j = 1, ...n + 1 \tag{C36}$$

in the regime under consideration. As we shall see, the boundary strings corresponding to $j \leq n$ in (C36) are breather boundary strings whereas, the boundary string of the highest order $\lambda_{(n)}$ corresponding to $j = n + 1$ in (C36) is

a close boundary string. By adding the boundary string $\lambda_{(0)}$ to the state $|1\rangle$ ($|-1\rangle$), we obtain the state $|1\rangle_{+B0}$ ($|-1\rangle_{+B0}$)(C20). Now one can add the higher order boundary string solution $\lambda_{(1)}$ to the state $|1\rangle_{+B0}$ ($|-1\rangle_{+B0}$). Let us consider the case $n > 1$. The boundary string $\lambda_{(1)}$ is a breather boundary string, and hence the charge of the state obtained by adding this boundary string $\lambda_{(1)}$ to the state $|1\rangle_{+B0}$ ($|-1\rangle_{+B0}$) has same charge as that of the state $|1\rangle_{+B0}$ ($|-1\rangle_{+B0}$). Let us denote this state which includes both the boundary strings $\lambda_0$ and $\lambda_{(1)}$ on top of the state $|1\rangle$ ($|-1\rangle$) as $|1\rangle_{+B1}$ ($|-1\rangle_{+B1}$). One can see that this process continues and by adding $k$ higher order boundary strings $\lambda_{(0)}, \lambda_{(1)}, ..., \lambda_{(k-1)}$, where $1 \leq k < n$ to the state $|1\rangle$ ($|-1\rangle$), we obtain a state whose total charge is equal to that of the state $|\pm 1\rangle$. We denote this state by $|1\rangle_{+Bk-1}$ ($|-1\rangle_{+Bk-1}$), and the associated density distribution by $\rho_{BBS,k-1}(\alpha)$. Note that due to the double pole, there exist degenerate states corresponding to the anti-symmetric superposition which we label $|1\rangle_{-Bk-1}$ ($|-1\rangle_{-Bk-1}$). We have

$$2m_0 \cosh \alpha - \frac{1}{L}\varphi'(\alpha, u) - 4\pi\rho_{BBS,k-1}(\alpha) - \frac{1}{L}\delta(\alpha) + \frac{1}{L}\varphi'(\alpha, u+\pi) - \frac{1}{L}\varphi'(\alpha, 3u) - \frac{1}{L}\varphi'(\alpha, u)$$
$$= \sum_{j=0}^{k-1} \frac{1}{L}\varphi'\left(\alpha, (2j+3)u\right) - \frac{1}{L}\varphi'\left(\alpha(2j-1)u\right) + \sum_\sigma \int \varphi'(\alpha + \sigma\gamma, 2u)\rho_{BBS,k-1}(\gamma)d\gamma. \quad \text{(C37)}$$

Now consider adding the boundary string $\lambda_{(k)}$ to the state $|1\rangle_{+Bk-1}$ ($|-1\rangle_{+Bk-1}$). We obtain the state $|1\rangle_{+Bk}$ ($|-1\rangle_{+Bk}$) described by the density distribution $\rho_{BBSk}(\alpha)$ which is the solution to the equation

$$2m_0 \cosh \alpha - \frac{1}{L}\varphi'(\alpha, u) - 4\pi\rho_{BBS,k}(\alpha) - \frac{1}{L}\delta(\alpha) + \frac{1}{L}\varphi'(\alpha, u+\pi) - \frac{1}{L}\varphi'(\alpha, 3u) - \frac{1}{L}\varphi'(\alpha, u)$$
$$= \sum_{j=0}^{k} \frac{1}{L}\varphi'\left(\alpha, (2j+3)u\right) - \frac{1}{L}\varphi'\left(\alpha, (2j-1)u\right) + \sum_\sigma \int \varphi'(\alpha + \sigma\gamma, 2u)\rho_{BBS,k}(\gamma)d\gamma. \quad \text{(C38)}$$

Since by adding the boundary string $\lambda_{(k)}$ to the state $|1\rangle_{+Bk-1}$ ($|-1\rangle_{+Bk-1}$), described by the density distribution $\rho_{BBS,k-1}(\alpha)$, we obtain the state $|1\rangle_{+Bk}$ ($|-1\rangle_{+Bk}$) described by the density distribution $\rho_{BBS,k}(\alpha)$, just as before, the difference in the density distributions $\rho_{BBS,k}(\alpha)$ and $\rho_{BBS,k-1}(\alpha)$ is just due to the boundary string $\lambda_{(k)}$, which we denote by $\Delta\rho_{BBS,k}(\alpha)$. we then have

$$\rho_{Bk}(\alpha) = \rho_{Bk-1}(\alpha) + \Delta\rho_{BBS,k}(\alpha). \quad \text{(C39)}$$

Using this in the equation (C38) and subtracting it from equation (C37), and applying the Fourier transform

$$\tilde{\varphi}'(\omega, y) = \int_{-\infty}^{\infty} dx \, e^{ix\omega}\varphi'(x, y) = 2\pi\frac{\sinh((\pi - (y - 2k\pi))\omega)}{\sinh(\pi\omega)}, \quad (2k\pi < y < (2k+2)\pi), \quad \text{(C40)}$$

to the resulting equation, we obtain

$$\tilde{\rho}_{BBS,k}(\omega) = \tilde{\rho}_{BBS,k-1}(\omega) + \Delta\tilde{\rho}_{BBS,k}(\omega), \quad \Delta\tilde{\rho}_{BBS,k}(\omega) = -\frac{1}{L}\frac{\cosh((2n+1)(\pi - u)\omega)\cosh((\pi - u)\omega)}{\cosh(u\omega)}. \quad \text{(C41)}$$

By iterating the above equation, and recalling that the density distribution of the state which contains no boundary strings is given by $\rho_{\pm 1}(\alpha)$ (C3), one can obtain the density distribution of all the states $|\pm 1\rangle_{\pm Bk}$, $k \leq n - 1$. We have

$$\tilde{\rho}_{BBS,k}(\omega) = \tilde{\rho}_{|\pm 1\rangle}(\omega) - \sum_{j=0}^{k} \frac{1}{L}\frac{\cosh((2j+1)(\pi - u)\omega)\cosh((\pi - u)\omega)}{\cosh(u\omega)}. \quad \text{(C42)}$$

Using the explicit form of the density distribution $\rho_{\pm 1}(\alpha)$ associated with the state $|\pm 1\rangle$ (C6) in the above equation, one can obtain the explicit expression for the density distribution $\rho_{BBS,k}(\alpha)$. The charge of this state containing $k+1$ ($k \leq n - 1$) boundary strings is

$$k + 1 + L\tilde{\rho}_{BBS,k}(0) = L\tilde{\rho}_{|\pm 1\rangle}(0) \quad \text{(C43)}$$

Where '$k+1$' on the left hand side corresponds to the $k+1$ boundary strings. Hence, all the states $|1\rangle_{+Bk}$ ($|1\rangle_{+Bk}$) obtained by adding the boundary strings $\lambda_{(0)}, ...\lambda_{(k)}$, $k \leq n - 1$ to the state $|1\rangle$ ($|-1\rangle$) have total normal ordered charge equal to that of the state $|\pm1\rangle$. Which proves the claim that all the boundary strings $\lambda_{(k)}$, $k \leq n - 1$ are breather boundary strings.

Now let us calculate the energy of the breather boundary string $\lambda_{(k)}$, $k \leq n - 1$. We have

$$E_{BBSk} = -m_0 \cosh(i(2k + 1)u) - m_0 L \int_{-\Lambda}^{\Lambda} d\alpha \, \cosh\alpha \, \Delta\rho_{BBS,k}(\alpha). \tag{C44}$$

This can be written as

$$E_{BBSk} = m_0 L \int_{-\Lambda}^{\Lambda} d\alpha \, \cosh\alpha \, \left( \frac{1}{2L} \sum_{\sigma=\pm} \delta(\alpha + \sigma i((2k+1)\pi - (2m+1)u)) - \Delta\rho_{BBS,k}(\alpha) \right). \tag{C45}$$

This can be expressed in terms of the Fourier transforms of $\cosh\alpha$ and the term in the parenthesis. We have

$$E_{BBSk} = \frac{m_0}{2\pi} \int_{-\infty}^{\infty} d\omega \, \frac{e^{\Lambda}}{2} \left( \frac{e^{i\Lambda\omega}}{1 + i\omega} + \frac{e^{-i\Lambda\omega}}{1 - i\omega} \right) \cosh\left((2n+1)(\pi - u)\omega\right) \frac{\cosh(u\omega) + \cosh((\pi - u)\omega)}{\cosh(u\omega)}. \tag{C46}$$

This can be evaluated in the same way as before. We obtain

$$E_{BBSk} = -2m \cos(\pi\gamma) \sin((2k+1)\pi/2) \sin((2k+1)\pi\gamma). \tag{C47}$$

The energy difference between the states $|\pm1\rangle_{\pm Bk}$ and $|\pm1\rangle$ is equal to the sum of the energies of the breather boundary strings $\lambda_{(0)}, ...\lambda_{(k)}$, $k \leq n - 1$, which is

$$E_{Bk} = \sum_{j=0}^{k} -2m \cos(\pi\gamma) \sin((2j+1)\pi/2) \sin((2j+1)\pi\gamma) = m \sin(k\pi\xi), \quad k = (1, ...n), \quad \xi = 2\gamma - 1, \quad n = [1/2\xi], \tag{C48}$$

where [...] denotes the integer part. One can check that $E_{Bk}$ is strictly positive. Now consider adding the boundary string of the highest order in this regime, which is $\lambda_{(n)} = \pm i(2n+1)u$ to the state $|1\rangle_{+Bn-1}$ ($|-1\rangle_{+Bn-1}$). Let us denote the density distribution of the resulting state by $\rho_{CBSn}(\alpha)$.

$$2m_0 \cosh\alpha - \frac{1}{L}\varphi'(\alpha, u) - 4\pi\rho_{CBSn}(\alpha) - \frac{1}{L}\delta(\alpha) + \frac{1}{L}\varphi'(\alpha, u + \pi) - \frac{1}{L}\varphi'(\alpha, 3u) - \frac{1}{L}\varphi'(\alpha, u)$$

$$= \sum_{j=0}^{n} \frac{1}{L}\varphi'(\alpha, (2j+3)u) - \frac{1}{L}\varphi'(\alpha, (2j-1)u) + \sum_{\sigma} \int \varphi'(\alpha + \sigma\gamma, 2u)\rho_{CBSn}(\gamma)d\gamma. \tag{C49}$$

Just as before, this equation can be solved by applying a Fourier transform. We obtain

$$\tilde\rho_{CBSn}(\omega) = \tilde\rho_{BBSn-1}(\omega) + \Delta\tilde\rho_{CBSn}(\omega), \quad \Delta\tilde\rho_{CBSn}(\omega) = -\frac{1}{L} \frac{\cosh((2n\pi - (2n+1)u)\omega)\sinh((\pi - 2u)\omega)}{\sinh((\pi\omega)) + \sinh((\pi - u)\omega)}. \tag{C50}$$

One can check that the normal ordered charge of the state obtained above by including all the allowed boundary strings (n breather boundary strings and the highest order boundary string) is zero. Hence this proves the claim that the boundary string $\lambda_{(n)}$ is a close boundary string, since it changes the normal ordered charge of the state $|1\rangle_{+Bn-1}$ ($|-1\rangle_{+Bn-1}$) to which it is added by an amount exactly equal to that of a soliton. We can calculate the energy of the close boundary string $\lambda_{(n)}$ just as before. We have

$$E_{CBSn} = -m_0 \cosh(2in\pi - i(2m+1)u) - m_0 L \int_{-\Lambda}^{\Lambda} d\alpha \, \cosh\alpha \, \Delta\rho_{CBSn}(\alpha). \tag{C51}$$

This can be written as

$$E_{CBSn} = m_0 L \int_{-\Lambda}^{\Lambda} d\alpha \, \cosh\alpha \, \left( \frac{1}{2L} \sum_{\sigma=\pm} \delta(\alpha + \sigma i(2n\pi - (2n+1)u)) - \Delta\rho_{CBSn}(\alpha) \right). \qquad (C52)$$

This can be expressed in terms of the Fourier transforms of $\cosh\alpha$ and term in the parenthesis. We have

$$E_{CBSn} = -\frac{m_0}{2\pi} \int_{-\infty}^{\infty} d\omega \, \frac{e^{\Lambda}}{2} \left( \frac{e^{i\Lambda\omega}}{1 + i\omega} + \frac{e^{-i\Lambda\omega}}{1 - i\omega} \right) \frac{\cosh((2n\pi - (2n+1)u)\omega)\sinh(\pi\omega)}{\cosh(u\omega)}. \qquad (C53)$$

This can be evaluated in the same way as before. We obtain

$$E_{CBSn} = -m \sin((2n+1)\pi/2) \sin(2n\pi\gamma) = -m \sin(n\pi\xi) \qquad (C54)$$

One can check that this is exactly equal and opposite to $E_{Bn-1}$, which is the sum of the energies of the breather boundary strings $\lambda_{(0)}, ...\lambda_{(n-1)}$ (C48). Hence by adding all the allowed boundary strings to the state $|1\rangle$ $(|-1\rangle)$, we obtain the state $|0\rangle_+$, which is degenerate with the states $|\pm 1\rangle$. Due to the double pole structure of the boundary strings, just as before, there exists a degenerate state $|0\rangle_-$. This proves that for all regimes of the values of $u$, there exist four fold degenerate ground states which are $|\pm 1\rangle, |0\rangle_\pm$.

#### d. Summary

The values of $u$ corresponding to different regimes lie in the range

$$(2n/2n+1)\pi < u < (2n+2/2n+3)\pi, \; n \in \mathbb{N}. \qquad (C55)$$

For $n = 0$ in the above equation, we obtain the regime $u < 2\pi/3$, where we have seen that there exists a unique boundary string solution $\lambda_{(0)} = \pm iu$, which corresponds to a close boundary string (has charge which is equal and opposite to that of a soliton) in that respective regime. When this boundary string is added to the state $|\pm 1\rangle$, one obtains the state $|0_+\rangle$ which is degenerate with $|\pm 1\rangle$. Due to the double pole associated with the boundary strings, there exists another degenerate state $|0\rangle_-$. As one increases $u$ such that it takes values in the regime $2\pi/3 < u < 4\pi/5$, which corresponds to $n = 1$ in the above equation (C55), the boundary string $\lambda_{(0)}$ becomes a breather boundary string (has zero charge), such that the state obtained by adding this boundary string $\lambda_{(0)}$ to the state $|1\rangle$ $(|-1\rangle)$ results in a state $|1\rangle_{+B0}$ $(|-1\rangle_{+B0})$, whose normal ordered charge is equal to that of $|\pm 1\rangle$, but has energy given by (C27) which is strictly positive. In the above states, the wavefunction associated with the boundary breathers is localized at both the boundaries in a symmetric superposition. Due to the double pole associated with the boundary strings, there exist degenerate states corresponding to the anti-symmetric superposition labeled by $|1\rangle_{-B0}$ $(|-1\rangle_{-B0})$. A new higher order boundary string $\lambda_{(1)} = \pm 3iu$ emerges, which falls into the close boundary string category. Adding this boundary string to the state $|1\rangle_{+B0}$ $(|-1\rangle_{+B0})$, one obtains the ground states $|0\rangle_\pm$. This occurs because the energy of the close boundary string $\lambda_{(1)}$ exactly cancels with that of the breather boundary string $\lambda_{(0)}$.

As we further increase the value of $u$ such that it takes values in the regime corresponding to $4\pi/5 < u < 6\pi/7$, which corresponds to $n = 2$ in the equation above (C55), the breather boundary string $\lambda_{(0)}$ remains as such, but the boundary string $\lambda_{(1)}$ which was a close boundary string in the previous regime now turns into a breather boundary string. Adding the boundary string $\lambda_{(1)}$ to the state $|1\rangle_{+B0}$ $(|-1\rangle_{+B0})$ now results in a state $|1\rangle_{+B1}$ $(|-1\rangle_{+B1})$, whose charge is exactly equal to that of the states $|\pm 1\rangle_{\pm B0}$ but its energy is higher than $|\pm 1\rangle_{\pm B0}$. A new higher order boundary string $\lambda_{(2)} = \pm 5iu$ arises, which is a close boundary string in the current regime. Adding this boundary string to the state $|1\rangle_{+B1}$ $(|-1\rangle_{+B1})$ results in the ground state $|0\rangle_\pm$. This occurs because the energy of the close boundary string $\lambda_{(2)}$ is exactly equal and opposite to the sum of the energies of the two breather boundary strings $\lambda_{(0)}$ and $\lambda_{(1)}$.

This process of appearance of higher order boundary strings continues as we increase the value of $u$ such that $n$ increases in (C55). Consider the regime corresponding to a certain $n > 1$ in (C55). There exist $n + 1$ boundary strings, where the boundary strings $\lambda_{(j-1)} = \pm i(2j-1)u$, $1 \leq j \leq n$ are breather boundary strings whereas, the boundary string of the highest order $\lambda_{(n)} = \pm i(2n-1)u$ is a close boundary string. By adding the boundary string $\lambda_{(0)}$ to the state $|1\rangle$ $(|-1\rangle)$, we obtain the state $|1\rangle_{+B0}$ $(|-1\rangle_{+B0})$. Now one can add the higher order boundary string

solution $\lambda_{(1)}$ to the state $|1\rangle_{+B0}$ ($|-1\rangle_{+B0}$). One then obtains the state $|1\rangle_{+B1}$ ($|1\rangle_{+B1}$) whose charge is exactly equal to that of the states $|1\rangle_{\pm B0}$ ($|-1\rangle_{\pm B0}$). Similarly, by adding the boundary strings $\lambda_{(0)}, \lambda_{(1)}, ..., \lambda_{(k-1)}$, where $1 \leq k < n+1$ to the state $|1\rangle$ ($|-1\rangle$), we obtain a state whose total charge is equal to that of the state $|\pm 1\rangle$. We denote this state by $|1\rangle_{+Bk-1}$ ($|-1\rangle_{+Bk-1}$). Again due to the double pole, there exist degenerate states corresponding to the anti-symmetric superposition labeled by $|1\rangle_{-Bk-1}$ ($|-1\rangle_{-Bk-1}$). The energy difference between the states $|\pm 1\rangle_{\pm Bk}$ and $|\pm 1\rangle$ is given by $E_{Bk}$ (C48), which is strictly positive, and hence these states are excited states. By adding the boundary string of the highest order in this regime, which is the close boundary string $\lambda_{(n)} = \pm i(2n+1)u$ to either of the states $|\pm 1\rangle_{+Bn-1}$, one obtains the states $|0_\pm\rangle$, which are degenerate with the states $|\pm 1\rangle$. The four states $|\pm 1\rangle, |0\rangle_\pm$ are the four degenerate ground states for all values of $u$, and the states $|\pm 1\rangle_{\pm Bk}$, $k \leq n-1$ are excited states on top of the ground states $|\pm 1\rangle$ in the current regime. As we increase the values of $u$, the number of allowed boundary strings $n+1$ increases, as $n$ increases, and as we approach the semi-classical limit $u \to \pi$ ($n \to \infty$), the number of allowed boundary strings become infinite. In this limit, the system still exhibits four fold degenerate ground state but since $E_{Bk} \to 0$ as $u \to \pi$, the excited states have arbitrarily small energies and eventually both the bulk and the boundary excitations become gapless when $u \sim \pi$.

### 3. Boundary Excitations

Having constructed the ground states $|\pm 1\rangle$ and $|0\rangle_\pm$, in this section we analyze the boundary excitations on top of each of these ground states. As we saw in the previous section, in the process of construction of the states $|0\rangle_\pm$, we obtained the boundary excitations on top of the states $|\pm 1\rangle$, which are $|\pm 1\rangle_{\pm Bk}$. To construct the excitations on top of the ground states $|0\rangle_\pm$, it is instructive to consider small boundary fields $\epsilon_{L,R} \sim 0$ at the left and right boundaries.

#### a. Bethe equations with small boundary fields

In the presence of the boundary fields $\epsilon_{L,R}$ corresponding to the left and right boundaries respectively, the integrable boundary conditions take the form

$$\Psi_L(0) = e^{i\phi}\Psi_R(0), \quad \Psi_L(L) = -e^{-i\phi'}\Psi_R(L), \tag{C56}$$

with

$$\text{OBC}: \phi = \epsilon_L + u - \pi/2, \ \ \phi' = \epsilon_R + u - \pi/2. \tag{C57}$$

Note that when $\epsilon_{L,R} = 0$, we recover the OBC (A3). In the presence of these boundary fields, the Bethe equations (A21) now take the form

$$e^{2im_0 L \sinh \beta_i} = \frac{\cosh\left(\frac{1}{2}(\beta_i + i\epsilon_L + iu)\right)}{\cosh\left(\frac{1}{2}(\beta_i - i\epsilon_L - iu)\right)} \frac{\cosh\left(\frac{1}{2}(\beta_i + i\epsilon_R + iu)\right)}{\cosh\left(\frac{1}{2}(\beta_i - i\epsilon_R - iu)\right)} \prod_{i \neq j, j=1}^{N} \frac{\sinh\left(\frac{1}{2}(\beta_i - \beta_j + 2iu)\right)}{\sinh\left(\frac{1}{2}(\beta_i - \beta_j - 2iu)\right)} \frac{\sinh\left(\frac{1}{2}(\beta_i + \beta_j + 2iu)\right)}{\sinh\left(\frac{1}{2}(\beta_i + \beta_j - 2iu)\right)}. \tag{C58}$$

The boundary conditions (C57) break the charge conjugation symmetry (A24), due to which the Bethe equations corresponding to the charge conjugated fermions differ from those of the regular fermions (C58). They are obtained by applying the transformation $\epsilon_{L,R} \to -\epsilon_{L,R}$ to (C58).

#### b. Structure of the boundary string solutions

Consider the Bethe equations corresponding to the charge conjugated fermions. Due to the boundary fields, they exhibit the following fundamental boundary string solutions $\lambda_{(0)}^L = \pm i(u - \epsilon_L)$, $\lambda_{(0)}^R = \pm i(u - \epsilon_R)$ corresponding to the left and right boundaries respectively. In the limit where the boundary fields vanish, the above two solutions merge where they correspond to the double pole, and as mentioned before, they are associated with bound states localized at both boundaries in a symmetric and anti-symmetric superpositions. For a certain $n$ in (C55), in the limit where the boundary fields $\epsilon_{L,R} \to 0$, by including the fundamental boundary string $\lambda_{(0)}^L$ corresponding to the left

boundary as one of the roots, one finds that there exists a higher order boundary string solution $\lambda_{(1)}^{L} = \pm i(3u - \epsilon_L)$. By including $\lambda_{(1)}^{L}$ as a root, a new higher order boundary string $\lambda_{(2)}^{L} = \pm i(5u - \epsilon_L)$ emerges. This continues until one reaches the highest order boundary string solution possible, which is $\lambda_{(n)}^{L} = \pm i((2n+1)u - \epsilon_L)$. The boundary strings $\lambda_{(j-1)}^{L}$, $j = 1, ..., n+1$ described above form a branch, which we call the first branch. In addition to the boundary strings corresponding to the first branch, there exists another set of boundary string solutions corresponding to another branch, which we call second branch to be discussed below. When the fundamental boundary string solution is included, in addition to the higher order boundary string $\lambda_{(1)}^{L}$ mentioned above, there exists another solution $\lambda_{(0)}^{'L} = \pm i(u + \epsilon_L)$. When this is included as one of the roots, a new higher order boundary string $\lambda_{(1)}^{'L} = \pm i(3u + \epsilon_L)$ emerges. This continues until one reaches the highest order boundary string solution corresponding to this branch, which is $\lambda_{(n)}^{'L} = \pm i((2n+1)u + \epsilon_L)$. Hence the second branch consists of boundary strings $\lambda_{(j-1)}^{'L}$, $j = 1, ...n+1$. Notice that the boundary string corresponding to the first branch $\lambda_{(j-1)}^{L}$ is related to that corresponding to the second branch $\lambda_{(j-1)}^{'L}$ through the transformation $\epsilon_L \rightarrow -\epsilon_L$.

Note that the Bethe equations corresponding to the original fermions are related to that of the charge conjugated fermions through the transformation $\epsilon_{L,R} \rightarrow -\epsilon_{L,R}$. Due to this, the boundary string solutions of the original fermions corresponding to the first and the second branches are identified with the second and the first branches respectively of the charge conjugated fermions described above. In the limit where the boundary field at the left boundary vanishes $\epsilon_L = 0$, the boundary string solutions corresponding to the two branches merge giving rise to a double pole structure. Similar to the left boundary, there exist two branches of boundary strings solutions corresponding to the right boundary, which are given by $\lambda_{(j-1)}^{R} = \pm i((2j-1)u - \epsilon_R)$ and $\lambda_{(j-1)}^{'R} = \pm i((2j-1)u + \epsilon_R)$ $j = 1, ..., n+1$. Similar to the left boundary, in the limit where the boundary field at the right boundary vanishes $\epsilon_R = 0$, the boundary string solutions corresponding to the two branches merge giving rise to a double pole structure. At both boundaries, the first n boundary strings in each branch $\lambda_{(0)}^{\alpha}, \lambda_{(1)}^{\alpha}, ...\lambda_{(n-1)}^{\alpha}$ and $\lambda_{(0)}^{'\alpha}, \lambda_{(1)}^{'\alpha}, ...\lambda_{(n-1)}^{'\alpha}$ are breather boundary strings whereas, the highest order boundary strings $\lambda_{(n)}^{\alpha}$ and $\lambda_{(n)}^{'\alpha}$ are close boundary strings, where $\alpha = L, R$. Note that the above description is strictly valid in the limit $\epsilon_{L,R} \rightarrow 0$. For finite values of the boundary fields $\epsilon_{L,R}$, the range of the values of $u$ where the above described boundary strings exist differs.

### c. Construction of excited states at the boundaries

As mentioned before, the presence of the boundary fields $\epsilon_{L,R}$ break the charge conjugation symmetry explicitly, due to which the Bethe equations corresponding to the original and charge conjugated fermions differ. In addition, the range of the values of $u$ where the above described boundary strings exist differs. This leads to a profound effect on the construction of the eigenstates which goes beyond the scope of this paper, and will be dealt with in a future publication. Here we describe the construction of the eigenstates in the limit where $\epsilon_{L,R} \rightarrow 0$.

*Charge $\pm 1$ states:* To construct the charge $+1$ states, we need to consider the Bethe equations corresponding to the charge conjugated fermions. As mentioned before, they are obtained by applying the transformation $\epsilon_{L,R} \rightarrow -\epsilon_{L,R}$ to (C58). By considering all the solutions to these equations on the $i\pi$ line which form a dense set we obtain the state $|1\rangle$. In the regime corresponding to a certain $n$ in (C55), by adding $k+1$ breather boundary strings corresponding to the left boundary $\lambda_{(j-1)}^{L}$, $j = 1, ...k+1$ where $k \leq n-1$, one obtains the state $|1\rangle_{LBk}$, ($L$ in the subscript $LBK$ denotes that the breather excitations are localized at the left edge) whose charge is equal to $+1$ and its energy with respect to the state $|1\rangle$ is still given by $E_{Bk}$ (C48), since we are working in the limit $\epsilon_{L,R} \rightarrow 0$. Hence there exist $n$ excited states at the left boundary on top of the state $|1\rangle$, which are labeled by $|1\rangle_{LBk}$, $k = 0, ..., n-1$. Similarly, there exist excited states labeled by $|1\rangle_{RBk}$, which are obtained by adding $k+1$ breather boundary strings corresponding to the right boundary $\lambda_{(j-1)}^{R}$, $j = 1, ..., k+1$ where $k \leq n-1$. The charge and the energy of these states is exactly equal to $|1\rangle_{LBk}$. Note that the states described above contain boundary strings corresponding to the first branch associated with the charge conjugated fermions.

To construct the states with charge $-1$, we need to consider the Bethe equations corresponding to the original fermions. Just as in the previous case, by considering solutions to these Bethe equations on the $i\pi$ line which form a dense set, we obtain the state $|-1\rangle$. When $\epsilon_{L,R} > 0$, the state $|1\rangle$ has higher energy compared to the state $|-1\rangle$, which is the ground state. In the limit $\epsilon_{L,R} \rightarrow 0$ the two states $|\pm 1\rangle$ are degenerate. Similar to the previous case, by adding $k+1$ breather boundary strings corresponding to the first branch of the original fermions, which are given by $\lambda_{(j-1)}^{'L}$ $j = 1, ...k+1$ where $k \leq n-1$ (recall that the first and the second branches of the charge conjugated fermions are

identified with the second and the first branches of the original fermions respectively), one obtains the state $|-1\rangle_{LBk}$, $k = 1, ..., n - 1$. The energy of these states with respect to the state $|-1\rangle$ is again given by $E_{Bk}$ (C48). Similarly, there exist states labeled by $|-1\rangle_{RBk}$ which are obtained by adding $k + 1$ breather boundary strings corresponding to the right boundary $\lambda'^R_{(j-1)}$, $j = 1, ..., k + 1$ where $k \leq n - 1$. The charge and the energy of these states is exactly equal to $|-1\rangle_{LBk}$. Hence the states $|1\rangle_{LBk}$, $|1\rangle_{RBk}$, $k = 0, ..., n - 1$ are excited states on top of the state $|1\rangle$ and the states $|-1\rangle_{LBk}$, $|-1\rangle_{RBk}$, $k = 0, ..., n - 1$ are excited states on top of the state $|-1\rangle$. There exist excited states which contain boundary breathers localized at both the boundaries. These states are represented by $|1\rangle_{LBk,RBl}$ and $|-1\rangle_{LBk,RBl}$, where they correspond to states containing $k + 1$ breather boundary strings corresponding to the left boundary and $l + 1$ breather boundary strings corresponding to the right boundary on top of the ground states $|\pm 1\rangle$ respectively. These states have energies $E_{Bk} + E_{Bl}$.

*Charge 0 states:* Consider the state $|1\rangle_{LBn-1}$. By adding the highest order boundary string in the first branch, $\lambda^L_{(n)}$, which is a close boundary string, one obtains the state $|0\rangle_R$. This state has charge zero and is degenerate with the state $|1\rangle$. The subscript $R$ is used to denote the localization of the bound state in the case $\epsilon_{L,R} > 0$ (This bound state becomes the ZEM when $\epsilon_{L,R} = 0$). Similarly, by adding the boundary string in the first branch, $\lambda^R_{(n)}$, which is also a close boundary string to the state $|1\rangle_{RBn-1}$, one obtains the state $|0\rangle_L$. In the limit $\epsilon_{L,R} \to 0$, the two states $|0\rangle_{L,R}$ are degenerate with the states $|\pm 1\rangle$. It is important to note that in the limit $\epsilon_{L,R} \to 0$, the state $|0\rangle_R$ can be constructed by adding the close boundary string $\lambda'^R_{(n)}$ to the state $|-1\rangle_{RBn-1}$ and similarly the state $|0\rangle_L$ can be constructed by adding the close boundary string $\lambda'^L_{(n)}$ to the state $|-1\rangle_{LBn-1}$.

Under charge conjugation, we have $\mathbb{C}|-1\rangle = |1\rangle$, and since the first and the second branches at each boundary transform into each other, we have $\mathbb{C}|1\rangle_{LBn-1} = |-1\rangle_{LBn-1}$. As mentioned above, by adding the close boundary string $\lambda^L_{(n)}$ to the state $|1\rangle_{LBn-1}$ we obtain the state $|0\rangle_R$ and by adding the close boundary string $\lambda'^L_{(n)}$ to the state $|-1\rangle_{LBn-1}$ we obtain the state $|0\rangle_L$. Hence we see that under charge conjugation we have $\mathbb{C}|0\rangle_L = |0\rangle_R$.

Up until now we have only added the boundary strings corresponding to the first branch. To obtain the excited states on top of the states $|0\rangle_{L,R}$, we need to consider the boundary strings corresponding to the second branch. Consider the state $|0\rangle_R$. By adding $k + 1$ breather boundary strings $\lambda'^L_{(j-1)}$, $j = 1, ..., k + 1$, $k \leq n - 1$ corresponding to the second branch associated with the left boundary, one obtains the state $|0\rangle_{R,LBk}$ which has total charge zero and its energy with respect to the state $|0\rangle_R$ is $E_{Bk}$ (C48). The letter $L$ in the subscript $LBK$ denotes that the breather excitations are localized at the left edge. There exist excited states on top of the ground state $|0\rangle_R$ which host breathers localized at the right edge, which are labeled by $|0\rangle_{R,RBk}$. They are constructed by adding $k + 1$ breather boundary strings $\lambda^R_{(j-1)}$, $j = 1, ..., k + 1$, $k \leq n - 1$ corresponding to the first branch associated with the right boundary. The state $|0\rangle_{R,RBk}$ has total charge zero and its energy with respect to the state $|0\rangle_R$ is $E_{Bk}$ (C48), and hence the states $|0\rangle_{R,LBk}$ and $|0\rangle_{R,RBk}$ are degenerate.

Similarly, by considering the state $|0\rangle_L$ and by adding $k + 1$ breather boundary strings $\lambda'^R_{(j-1)}$, $j = 1, ..., k + 1$, $k \leq n - 1$ corresponding to the second branch associated with the right boundary, one obtains the state $|0\rangle_{L,RBk}$ which has total charge zero and its energy with respect to the state $|0\rangle_L$ is $E_{Bk}$ (C48). They host breathers localized at the right boundary which is represented by the letter $R$ in the subscript $RBK$. There exist excited states on top of the ground state $|0\rangle_L$ which host breathers localized at the left boundary which are labeled by $|0\rangle_{L,LBk}$. These states are constructed by adding the $k + 1$ breather boundary strings $\lambda^L_{(j-1)}$, $j = 1, ..., k + 1$, $k \leq n - 1$ corresponding to the first branch associated with the left boundary to the state $|0\rangle_L$. They have zero total charge and are degenerate with $|0\rangle_{L,RBk}$.

Hence the states $|0\rangle_{L,LBk}$, $|0\rangle_{L,RBk}$, $k = 0, ..., n - 1$ are excited states on top of the state $|0\rangle_L$ and the states $|0\rangle_{R,RBk}$, $|0\rangle_{R,LBk}$, $k = 0, ..., n - 1$ are excited states on top of the state $|0\rangle_R$. These states are degenerate with the states $|\pm 1\rangle_{LBk}$, $|\pm 1\rangle_{RBk}$, $k = 0, ..., n - 1$, which are excited states on top of the states $|\pm 1\rangle$. Unlike in the case of $|\pm 1\rangle_{\alpha Bn-1}$, $\alpha = L, R$, in order to add the highest boundary string to either of the states $|0\rangle_{\alpha,LBn-1}$, $|0\rangle_{\alpha,RBn-1}$, $\alpha = L, R$, one needs to add a soliton. There exist excited states which contain boundary breathers localized at both the boundaries. These states are represented by $|0\rangle_{L,LBk,RBl}$ and $|0\rangle_{R,LBk,RBl}$, where they correspond to states containing $k + 1$ breather boundary strings corresponding to the left boundary and $l + 1$ breather boundary strings corresponding to the right boundary on top of the ground states $|0\rangle_{L,R}$ respectively. These states have energies $E_{Bk} + E_{Bl}$.

When the boundary fields vanish $\epsilon_{L,R} = 0$, the Bethe equations corresponding to the charge conjugated and original fermions are exactly the same. The boundary strings corresponding to the left and the right boundaries merge giving

rise to double poles in the Bethe equations, and in addition, the boundary strings corresponding to the first and the second branch associated with each boundary also merge. We label the boundary strings at both the left and right boundaries corresponding to one branch as $\lambda_{(j)}$ ($\lambda_{(j)}^L = \lambda_{(j)}^R \equiv \lambda_{(j)}$), whereas the boundary strings corresponding to the other branch are labeled as $\lambda_{(j)}'$ ($\lambda_{(j)}'^L = \lambda_{(j)}'^R \equiv \lambda_{(j)}'$). In this limit of vanishing boundary fields, although the four different boundary string solutions $\lambda_{(j)}^L, \lambda_{(j)}^R, \lambda_{(j)}'^L, \lambda_{(j)}'^R$ take the same form $\pm i(2j+1)u$, the number of eigenstates associated with each boundary string solution however remains unchanged. In this case of vanishing boundary fields, as mentioned previously, boundary strings describe bound states that are localized at both the boundaries in a symmetric and anti-symmetric superpositions. The construction of the excited states described above for the case of vanishing boundary fields is summarized in the main text.

Note that we could have added the higher order boundary strings $\lambda_{(l)}'^L$, $0 \leq l < n-1$ corresponding to the second branch associated with the left boundary to the state $|1\rangle_{Bk}$, for any $0 \leq k < n-1$. This results in more states which are not shown in the previous construction, whose total charge is same as that of the state $|1\rangle$ and have energies $E_{Bl} + E_{Bk}$, $0 \leq k, l \leq n-1$ with respect to the state $|1\rangle$. Similar construction exists at the right boundary. However, it was argued in [33], that these states are not physical as they do not correspond to any poles in the physical boundary $S$-matrix. It might be instructive to investigate the 'non-physical' nature of these states.