# Peer review of "Duality symmetry, zero energy modes and boundary spectrum of the sine-Gordon/massive Thirring model"

_SciPost Physics_

## Round 1 · Referee Report · Anonymous (Referee 1) · 2025-8-7

Strengths

In this paper, the authors utilise the integrability of the massive Thirring / sine-Gordon model in order to study the spectrum on the boundary. They find a topological phase with a ground state degeneracy in the presence of a boundary along with a trivial phase without this degeneracy. Furthermore, they find a sub-gap boundary spectrum for values of parameters where interactions between the solitons are attractive. Furthermore, they discuss a duality between parameters of the model and boundary conditions.

This is a very technical paper, which is completely expected for work exactly solving a strongly correlated quantum model. Many of the questions in the field are quite subtle involving e.g. stability under quantum fluctuations of the edge states under general parameter regimes (it has been proven previously for some portions of the phase diagram but not all), and to the best of my knowledge, this is the first time the boundary breather spectrum has been calculated. This work is certainly of great interest to people in the field.

Weaknesses

The manuscipt in its present form is rather unclear, particularly in regard to what the new results are.

Report

I find the paper in its present form rather difficult to follow. The introduction and abstract do not make clear at all what are the outstanding problems solved in this paper, and throughout the manuscript, the distinction between previously known results and new ones are blurred. Furthermore, while it makes sense to put details of the calculation in appendices, I found there is not enough information in the main text to follow it without first reading the appendices carefully.

I would suggest a medium-scale edit of the manuscript for overall clarity, including (but not limited to) the following points:

1a. Almost the entirety of the abstract is about the duality between the two different boundary conditions. This is almost trivial in the sine-Gordon formulation (just let beta Phi -> beta Phi + pi in Eq. 1) and has been discussed before (including e.g. in Ref. 24 by some of the same authors). It may be slightly less trivial in the fermonic formulation, but the authors don’t devote much discussion to any potential (and subtle!) differences between the sine-Gordon and massive Thirring models. There is no problem with mentioning this duality, but I do not believe it to be a significant result of this work, unless the authors can give a convincing reason of why one would not have expected it to hold.

1b. On a similar note, it would be very useful throughout the paper to make the decision either to treat one boundary condition and let m_0 be positive or negative, or (probably the better choice for this work) to keep m_0 positive and deal with the two different boundary conditions. One important example where a fixed convention would have been helpful is the difference between Eqs. A21 and A23.

  1. The work is motivated by 1D spin-triplet superconductors. However these also have a charge sector, which is important for physical properties. There is no problem with this motivation, but it would be useful to then be clear afterwards to say that this work concentrates on the technical aspect of analysing just the sine-Gordon model. It may also be give some other potential realisations of the sine-Gordon model, such as the XXZ model (which results are also widely compared to in the integrability literature), but there are many other possibilities.

3a. The introduction could have had a much more comprehensive analysis of what is known about the quantum sine-Gordon model (with Dirichlet boundary) — there is a lot of previous literature on the idea that there is a trivial phase for one sign of m_0 and a topological phase with edge states for the other sign. While much of this relies on a classical analysis, there is other work (including Ref. 24 by some of the same authors) studying this from the point of view of integrability. It would have been very useful at the beginning to know what is similar about this analysis and where it differs (and more precisely where it extends previous results). Many of these points are subtle, so may be missed by non-expert readers who may fail to understand what is new in this work.

3b. As a paper on the boundary sine-Gordon model, it seems odd not to reference the first works in the field of Ghoshal and Zamolodchikov (1994) and Ghoshal (1994), even in passing.

  1. There is no problem with putting full details in the appendix, even where the follow from previous work as it sets notation and makes the manuscript (reasonably) self-contained. However it would be very useful for the reader to be more clear on what is following previous work and what are the new results. For example, Eqs A21 and A23 (and 20 in the main text) are basically the same as Eq. 3.12 in Skorik and Saleur (Ref 33). The calculation does follow Skorik and Saleur further than this, but I didn’t follow this in full detail. The authors of this paper should be much more clear on this though.

5a. I would suggest putting some of the key equations from the appendix into the main text. It is far from obvious from Eq. 20 why m_0<0 and m_0>0 have very different solutions, or what boundary strings are. As this is a technical paper and results are subtle, some of these details are important, particularly the ones that are new in this work and can be highlighted. I understand that this isn’t an easy task, but I would suggest as a minimum putting in the main text enough to see why boundary states appear in one case and not the other, and what the boundary strings are.

5b. I would also suggest in the many occasions where the reader is referred to ‘the appendix’ that the authors are more specific as there are multiple appendices which are quite lengthy.

Requested changes

Edit for clarity -- see report for some specific suggestions.

Recommendation

Ask for major revision

---

## Round 1 · Referee Report · Anonymous (Referee 2) · 2025-8-21

Report

The authors study the energy spectrum of the SGM/MTM with Dirichlet boundary conditions. In practice, this is done using Bethe ansatz on the MTM. They obtain the trivial and topological phases, distinguished by the ground-state degeneracy embedded in the discrete symmetries. In principle the results seem valuable for publication in SciPost Phys (Core), but, as also pointed out by the 1st referee, the presentation should be clarified. In particular, I would like to ask (i) to give more details on the appearance of the boundary anomaly in (11), (ii) add details on the ground state in the trivial case (ie, the Bethe roots) like done in the topological case, and (iii) point out more clearly which appendices are relevant at which point. In addition, I believe in the end the findings should be discussed (ie, translated back) in the context of the SGM and linked to the boundary formalism introduced by Ghoshal and Zamolodchikov.

Requested changes

see report

Recommendation

Ask for major revision

---

## Round 1 · Referee Report · Anonymous (Referee 3) · 2025-8-22

Strengths

  1. Provides an exact construction of the ground state manifold of the model.
  2. Constructs the spectrum of boundary excitations.
  3. Unveils an interesting sequence of solutions to the Bethe equations.

Weaknesses

  1. The structure of the paper make it difficult to read, as do the numerous typos and inconsistent notation.
  2. Some important points, suchs as the "boundary anomaly" are not explained.
  3. The full range of parameters is not explored.

Report

In this paper the authors investigate the ground state manifold of the massive Thirring model in the presence of particular boundary conditions. They find that in certain parameter regimes the model exhibits a 4-fold ground state degeneracy which is associated to the presence of zero energy boundary bound states. A series of boundary excitations on top of these ground states are then also constructed. This is achieved by using the exact solution of the model and unveiling an intricate sequence of solutions to the Bethe Ansatz equations which give rise to the zero energy modes. The results are interesting and the method by which the spectrum is constructed would certainly be of interest to those working in integrability and boundary physics.

It is my opinion, however, that in current form the manuscript is not fit for publication. It is structured in a way that makes it difficult to read and is filled with typos and mixed notation. Moreover many important points are not clearly explained. In particular, it is structured somewhat in the style of a letter with many details presented in the appendices. Since space is not an issue I suggest a major restructuring of the manuscript with minimal use of appendices and systematic approach to the solution of the model and the regimes. This could be supplemented by a section in the introduction, briefly previewing the results. Below are more detailed points on the content of the paper.

Requested changes

  1. The introduction is lacking in both citations to previous works and explanation of what this paper adds. Several seminal works on the sine gordon with boundary fields are not mentioned e.g. Ghoshal & Zamolodchikov arXiv: 9306002 or Caux et. al arXiv: 0306328. A similar model was considered by the some of the authors in references 24 and 25, how does this paper differ, in approach techniques or results.

  2. In the abstract the authors state “We solve the…”, however the solution seems to already appear in reference 33, indeed the Bethe equations appear there already. I suggest the authors modify this sentence and the introduction to account for this. Moreover, the authors state “We provide the exact solution of the SG model… for all values of \beta”. This statement is somewhat misleading since later they restrict their analysis to a particular range of values ion the interaction parameter u. While the Bethe equations are the same for any value of the interaction the construction of the ground state is not. I suggest the authors rephrase this sentence.

  3. On the same point, it was shown how to construct the ground state in the regime not considered by the authors in “The mass spectrum and the S matrix of the massive Thirring model in the repulsive case” by Korepin. Perhaps the authors can use this to provide a complete analysis in all regimes following on from this paper.

  4. Below equation 5 it is stated that the chiral symmetry \Psi_R <-> \Psi_ is broken by the mass. Performing this transformation in (4) at g=0 the Hamiltonian gets mapped to itself with an overall minus sign, independent of the presence of the mass.

  5. Below equation (6) it is stated that charge conjugation is equivalent to \Phi-> -\Phi . Is this consistent with the definitions in (3) and (6). Perhaps there is a typo or more explanation is needed.

  6. Below (7), it would be useful to state what the restriction on u means in terms of beta. And if there is any physical meaning to this restriction or is it a byproduct of the Bethe ansatz basis?

  7. The boundary equations (10) and (11) are not the same as what would be naively implemented using bosonization/fermionization. This is referred to as an “boundary anomaly” however there is no explanation of this. What does this mean? In the masses limit, the model reduces to a simple Luttinger liquid which is very well studied, since the factor does not depend on the mass, does this anomaly appear there also? If so, how can one understand it from bosonization. The word anomaly in QFT has a specific meaning, is it related to that? It is stated that the boundary conditions preserve charge conjugation symmetry in the appendix, but the appendix just refers back to the main text without any explanation of why? I suggest that the authors add a clear discussion of these points.

  8. Throughout the text the authors draw a distinction between the solution for m_0>0 and m_0<0. For example equations 17 and 20 are stated to be the Bethe states and Bethe equations for m_0>0. However, these are the Bethe equations for arbitrary m_0 and u. What changes when m_0 is positive or negative is which solution of the Bethe equations is the ground state, e.g if m_0< 0 you can take \beta_j real if on the other hand m_0>0 you can take Im(\beta_j)= pi .

  9. In section V below equation 28, the notation \lambda_(0) is introduced but then followed by a definition of \lambda without the subscript. Is this a typo? Also there is a line break at the top of page 5 for no reason.

  10. Much of the notation and nomenclature in sections IV and V is unclear without reading the appendices first e.g. the phrase close boundary string. This can be helped by restructuring the paper and eliminating the appendices. Why are close boundary strings associated with exponentially decaying wave functions and is this not the case for boundary breathers.

  11. In figure 1 the notation is not consistent with the text and neither is the notation of the appendices. In (17) alpha denotes an index and beta a rapidity whereas in the figure alpha is a rapidity and beta does not appear. This mismatch also appears above equation 32 where {\alpha_k} are described as a set of be the roots.

  12. How are equations A11 and A12 derived?

  13. As in point 8 above: It is stated that A21 are the Bethe equations for m_0>0 and A23 are those for m_0<0. How can this be, there is no dependence on the sign of the mass in the derivation of the Bethe equations which arise form the quantisation condition of the wave function. A20, with A21 defines an eigenstate of the Hamiltonian for arbitrary m_0.

  14. Below A24 it is stated that the Hamiltonian is invariant under charge conservation however this would appear to be a subtle point because of the boundary conditions. It would be helpful if the authors could provide details of the Hamiltonian as written in (4) is invariant. Naively it would seem this requires some integration by parts which would then involve the appearance of a boundary term.

  15. In appendix B the authors carry out the bulk of their analysis in the thermodynamic limit. They follow the standard approach of the thermodynamic Bethe Ansatz however there are well known subtleties in this approach when periodic boundary conditions are not used, see e.g arXiv :1003.5542 . Perhaps the authors can comment on why they need not consider these extra boundary terms.

16.As in point 11 above: In C1 the notation has changed, alpha is now a rapidity instead of an index, In C.2 there is a typo in the argument of the \varphi function appearing in the sum on the right hand side.

  1. Everywhere the authors allow the boundary strings to be shifted by multiples of 2i pi, is it necessary to state this. Can’t you just say the you restrict to a certain range of imaginary parts?

  2. It is not obvious why the certain boundary strings and states correspond to symmetric or anti symmetric combinations of bound states at either edge. Could the authors expand upon this.

  3. It is stated that C13 is the charge of the CBS0 state, but this is not consistent with the definition B.14

  4. Below C.23 it is stated that boundary strings charge q, given the definition of the charge this should be 1.

  5. The appearance of \lambda_{(1)} as solutions to the Bethe equations only in the presence of \lambda_{(0)} which lies at the heart of the construction is merely stated but not explained. I suggest that the authors expand upon this.

  6. It can sometimes happen that solutions of the Bethe equations do not correspond to physical states. The main example of this is the the \beta=0 solution for the Dirichlet boundary conditions which results in a vanishing wave function. Have the authors checked that their intricate sequence of boundary states and excitations all correspond to physical states?

Recommendation

Ask for major revision

---

## Editorial Decision

awaiting_resubmission